# Green electrosynthesis of 3,3'-diamino-4,4'-azofurazan energetic materials coupled with energy-efficient hydrogen production over Pt-based catalysts

Jiachen Li[1], Yuqiang Ma[1], Cong Zhang[1], Chi Zhang[1], Huijun Ma[2], Zhaoqi Guo[1], Ning Liu[3], Ming Xu [4] ✉, Haixia Ma [1] ✉ & Jieshan Qiu[5] ✉

The broad employment of clean hydrogen through water electrolysis is restricted by large voltage requirement and energy consumption because of the sluggish anodic oxygen evolution reaction. Here we demonstrate a novel alternative oxidation reaction of green electrosynthesis of valuable 3,3'-diamino-4,4'-azofurazan energetic materials and coupled with hydrogen production. Such a strategy could greatly decrease the hazard from the traditional synthetic condition of 3,3'-diamino-4,4'-azofurazan and achieve low-cell-voltage hydrogen production on $WS_2$/Pt single-atom/nanoparticle catalyst. The assembled two-electrode electrolyzer could reach 10 and 100 mA cm$^{-2}$ with ultralow cell voltages of 1.26 and 1.55 V and electricity consumption of only 3.01 and 3.70 kWh per m$^3$ of $H_2$ in contrast of the conventional water electrolysis (~5 kWh per m$^3$). Density functional theory calculations combine with experimental design decipher the synergistic effect in $WS_2$/Pt for promoting Volmer–Tafel kinetic rate during alkaline hydrogen evolution reaction, while the oxidative-coupling of starting materials driven by free radical could be the underlying mechanism during the synthesis of 3,3'-diamino-4,4'-azofurazan. This work provides a promising avenue for the concurrent electrosynthesis of energetic materials and low-energy-consumption hydrogen production.

Excessive consumption of fossil fuels leads to the aggravating environmental pollution, and greenhouse effect. Hydrogen ($H_2$) is considered as one of the promising alternative energy carrier benefiting from its sustainable, zero-carbon emission, regenerative, and high intrinsic energy density ($\approx$282 kJ mol$^{-1}$) nature[1,2]. Electrochemical overall water splitting (OWS) powered by abundant solar or wind energy sources etc. represents a sustainable approach to produce $H_2$[3]. Yet, its implementation faces challenges of heavy electricity consumption (>4.5 kWh m$^{-3}$ $H_2$) due to the low-energy conversion efficiency and high operating voltage (>1.8 V), majorly originated from the sluggish four-electron transfer kinetics of oxygen evolution reaction (OER) in the anode[2,4,5]. In extreme cases, water electrolysis produces

[1]Xi'an Key Laboratory of Special Energy Materials, School of Chemical Engineering, Northwest University, Xi'an 710069, China. [2]National Demonstration Center for Experimental Chemistry Education, Northwest University, Xi'an 710127, China. [3]Xi'an Modern Chemistry Research Institute, Xi'an 710065, China. [4]State Key Laboratory of Chemical Resource Engineering, Beijing Advanced Innovation Center for Soft Matter Science and Engineering, Beijing University of Chemical Technology, Beijing 100029, China. [5]State Key Laboratory of Chemical Resource Engineering, College of Chemical Engineering, Beijing University of Chemical Technology, Beijing 100029, China. ✉e-mail: mingxu@mail.buct.edu.cn; mahx@nwu.edu.cn; qiujs@mail.buct.edu.cn

$H_2/O_2$ mixtures crossover through the membrane may give rise to gas explosive, causing considerable security risks[6]. In addition, the $O_2$ obtained from water electrolysis is relatively less valuable because it can be expediently obtained from air[7,8]. In parallel to the design of advanced electrocatalysts for OER, considerable research efforts are required to explore alternative technological concepts to surpass the conventional OWS technology and tackle the above drawbacks brought by OER[9].

Given the challenges in conventional OWS, efficient alternative strategies have been recently employed. Among them, the co-electrolysis system, appending different electrolytes in the anode and coupled with HER to integrate hybrid electrolysis, have witnessed increasing attention[10]. For instance, the HER paired with some thermodynamically more favorable electrochemical oxidation reactions such as hydrazine[11–13], urea[14,15], alcohols[1,16], aldehyde[17,18] etc., significantly lowering the required cell voltage of OWS for low-energy-consumption hydrogen production at the cathode. Meanwhile, the approach simultaneously provides either value-added fine chemicals or safe gas from pollutant degradation (e.g., $N_2$) at the counter side. Moreover, this co-electrolysis strategy could avoid $O_2$ generation and eliminate the risk of explosive $H_2/O_2$ mixtures, making highly-efficient membrane-less OWS reactors possible[19]. In view of this co-electrolysis strategy is still in the initial stage, intensive research should be conducted to explore more promising alternative electrooxidation reactions in the configuration of hybrid OWS system, particularly for the green and sustainable pathway to synthesize value-added chemicals, so that to expand the versatility and enhance the industrialization possibility of this strategy[8].

Furazan chemistry has proceed over 140 years, many derivatives have been synthesized and used for diverse applications, ranging from biological medicine to explosive and propellant ingredients[20]. Among them, the azofurazan structural motif exists in the form of crown ether analogs and considered as biologically active molecules suppressant of the soluble guanylate cyclase[20]. The azofurazan framework characterized with high-nitrogen and high heats of formation features have been most extensively researched as energetic materials (EMs) due to the large number of N–N and C–N bonds and the thermodynamically favored formation of $N_2$[21]. However, the traditional pathway for the synthesis of azofurazan EMs is an oxidation reaction from aminofurazans that often involves high-temperature synthetic condition and hazardous reagents such as $KMnO_4/H^+$, $CrO_3/AcOH$, $(NH_4)_2S_2O_8$ or various of organic reagents, which are a potential danger for the safety hazard and environmental pollution. In addition, the separation of product from excess of an oxidant and other by-products is difficult[20,21]. These drawbacks have compelled in a host of ecological concerns, the green synthetic techniques have been deserved great attention in the modern synthetic chemist. One such green technique, electrosynthesis has gone through a renaissance of sorts and considered an environmentally benign methodology in recent years[22–24]. The electron in electrosynthesis represents a clean and environmentally friendly reagent for direct oxidation, as it produces no waste instead of hazardous oxidizing agents. The electron offers greater economy in contrast to chemical redox reagents in view of the reagent cost per mole of product produced[25]. Moreover, mild synthetic condition is feasible because the energy supplied to the electrode could be controlled by the applied cell voltage and current density instead of the heat or high pressure. In terms of the advantages that the synthetically organic electrochemistry offers, many pioneer literatures have been reported detailing the utility of electro-synthetic organic chemicals through the coupling of C–H/X–H (X = C, O, S, N etc.) to construct C–C, C–O, C–S bonds etc[23,24]. Unfortunately, there has been a notable lack of researches on the electrosynthesis of organic EMs, especially for electrochemical N=N bond formations to generate azofurazan EMs[20,26]. More importantly, the formation of N=N bond was found thermodynamically more favorable than OER during the electrosynthesis of some azo EMs[27,28]. Therefore, it is speculated that the sluggish OER is replaced by electrosynthesis of azofurazan EMs and coupled with OWS may significantly reduce the cell voltage of OWS for low-energy-consumption hydrogen production, meanwhile, the OER is inhibited in the coupling system that could avoid many drawbacks brought by oxygen. Moreover, the azofurazan EMs are obtained at the anode through green and mild electrooxidation pathway.

With this idea in mind, a low-energy-consumption coupling system was constructed in our work by replacing OER with thermodynamically more favorable electrooxidation of 3,4-diaminofurazan (DAF) to synthesize 3,3'-diamino-4,4'-Azofurazan (DAAzF) EMs. The platinum single atoms/nanoparticles ($Pt_{1,n}$) supported on the tungsten disulfide nanosheets and loaded on the 3D conductive carbon cloth ($CC@WS_2/Pt_{1,n}$) was prepared as high-efficiency HER catalysts. The Pt nanoparticles (Pt NPs) loaded near the Pt single atoms (Pt SAs) may induce the variation of electronic metal-support interaction (EMSI) to regulate the charge redistribution between the neighboring Pt NPs/SAs and support, which was closely related to the catalytic activity[29]. It is noted that single atoms are not the only active sites in many single-atom catalysts, where the nanoclusters exist as well due to the extremely high surface free energy of single atoms, which is usually overlooked[30]. Hence, the formation of $Pt_{1,n}$ on the substrate is energetically more favored and more stable than pure Pt SAs under long-term harsh alkaline working condition[31]. The only single atoms active sites may also not exhibit their overwhelming advantages in multi-step catalytic process (e.g., alkaline HER)[32–34]. Compared with carbon-based substrates, the transition metal dichalcogenides (TMDs) supported metal single atoms/nanoparticles could adjust the electronic structure through both anchoring atom and the adjacent transition metal atoms with higher atomic number, which provided more flexible local coordination environment to regulate the catalytic activity[35]. The core anchoring chalcogen (S) and the adjacent transition metal (W) can synergistically regulate the $d$-orbital state electronic structure of metal single atoms/nanoparticles through EMSI. As a result, the adsorption energy of intermediates on active sites could be optimized and thus has influence on the catalytic activities[36]. As for one of the important candidates in TMDs, the metallic $WS_2$ achieved high catalytic activity for HER compared with other TMDs because the basal plane of metallic $WS_2$ was also catalytically active than natural $MoS_2$[37,38]. The periodic density functional theory calculations also indicated that the catalytic activity of $WS_2$ was predicted to be comparable to or even better than $MoS_2$[39]. The as-prepared $CC@WS_2/Pt_{1,n}$ can be directly used as self-supporting electrode for superior alkaline HER. DFT calculations revealed the facile energy barrier of water dissociation and H intermediates (H*) contributed by the synergistic effect among Pt NPs, Pt SAs, and $WS_2$ substrate, resulting in high alkaline HER performance. The as-prepared $CC@WS_2/Pt_{1,n}$ cathode integrated with anode electrode (copper oxide nanowires) to construct two-electrode coupling system, which can produce $H_2$ with an ultralow cell voltage in contrast to traditional OWS, realizing low-energy-consumption hydrogen production. Meanwhile, green electrosynthesis of DAAzF was achieved to avoid hazardous synthetic condition of traditional strategy (Fig. 1).

## Results

### Preparation and structural characterizations of cathodic $CC@WS_2/Pt_{1,n}$ catalyst

The cathode $CC@WS_2/Pt_{1,n}$ catalysts was synthesized based on the hydrothermal reaction to produce $WS_2$ nanosheets that was supported on the CC ($CC@WS_2$ NSs), followed by the electrochemical cyclic voltammetry (CV) method at a constant potential range to homogeneously distribute Pt NPs and Pt SAs ($Pt_{1,n}$) on the $CC@WS_2$ NSs (Fig. 2a). Scanning electron microscope (SEM) shows that dense of cross-linking $WS_2$ NSs are uniformly supported on the conductive CC substrate (Supplementary Fig. 1). Transmission electron microscopy (TEM), high-resolution TEM (HRTEM), and elemental mapping

indicated the uniform distribution of W and S elements on the WS$_2$ NSs (Supplementary Fig. 2). After the electrochemical deposition of Pt$_{1,n}$, negligible differences in morphology were observed in SEM indicating the sub-nanometer clusters or single atoms of Pt in WS$_2$/Pt$_{1,n}$ (Fig. 2b). Because the HER performance depends largely on the content of Pt$_{1,n}$, if not otherwise stated, the CC@WS$_2$/Pt$_{1,n}$ mentioned in the following sections was the optimal Pt content (2.0 mL Pt$^{4+}$, 10 mg mL$^{-1}$ in the electrolyte). To further verify the structure of Pt, aberration-corrected high-angle annular dark-field (AC-HAADF-STEM) was performed and shown in Fig. 2c, d. The prepared Pt with the nanoparticles and single atoms (within white dotted circle) forms were dispersed uniformly on the surface of corrugate WS$_2$ NSs. The average size of Pt NPs was determined to 2.6 nm (Supplementary Fig. 3), and the content of Pt$_{1,n}$ in WS$_2$/Pt$_{1,n}$ was 0.106 mg$_{Pt}$ cm$^{-2}$ as determined by the inductively coupled plasma mass spectrometer (ICP-MS). The average lattice spacing of 0.232 nm can be indexed to the Pt (111) plane, indicating the successfully synthesis of Pt NPs without any impurities (Fig. 2e)[40]. Besides, the elemental mapping of the WS$_2$/Pt$_{1,n}$ showed Pt$_{1,n}$ along with W and S were homogeneously distributed on the WS$_2$ NSs (Fig. 2f–i). Notably, the concordant coexistence of Pt NPs and Pt SAs on the WS$_2$ substrate was attributed to the strong coupling effect among these three active components in virtue of EMSI, which could synergistically promote alkaline HER (as discussed in the electrochemical tests and theoretical calculations section).

The X-ray diffraction (XRD) patterns showed a broad peak located at around 21.6° was indexed to CC substrate, the diffraction peaks around 14.3° and 32.7° can be assigned to (002) and (100) planes of hexagonal WS$_2$ (JCPDS no. 08–0237), respectively. After the electrodeposition of Pt$_{1,n}$ on the WS$_2$ NSs, two new peaks presented at around 39.8° and 46.2° were attributed to the metallic Pt (JCPDS no. 04–0802), which was well consistent with the TEM analysis and further supporting the successful synthesis of Pt NPs on the WS$_2$ NSs (Fig. 3a). We also investigated the impact of different Pt$_{1,n}$ content in CC@WS$_2$/Pt$_{1,n}$ toward XRD patterns (Supplementary Fig. 4). The content of Pt$_{1,n}$ in CC@WS$_2$/Pt$_{1,n}$ could be well controlled by adjusting the volume of Pt$^{4+}$ in electrolyte. No apparent diffraction peaks related to Pt were observed if the volume of Pt$^{4+}$ lower than 1.0 mL, suggesting the low content and small size of Pt on WS$_2$. The diffraction peak intensity increased with the volume of Pt$^{4+}$ up to 3.0 mL, indicating the high-loading and aggregation of Pt NPs (Supplementary Fig. 5). In order to further verify whether the phase transition of WS$_2$ occurred after the

deposition of Pt$_{1,n}$, the Raman spectra of CC@WS$_2$ and CC@WS$_2$/Pt$_{1,n}$ were performed. As shown in Fig. 3b, two strong features located at 352 and 416 cm$^{-1}$ in both CC@WS$_2$ and CC@WS$_2$/Pt$_{1,n}$ corresponded to $E^1_{2g}$ and $A_{1g}$ of 2H phase of WS$_2$, respectively[41]. Some weak peaks ($J_1$, $J_2$) at low frequency region were typically metastable 1 T phase of WS$_2$. Notably, the relative intensities of $A_{1g}/E^1_{2g}$ increased from 0.96 of CC@WS$_2$ to 1.04 of CC@WS$_2$/Pt$_{1,n}$, indicating the dopant of Pt$_{1,n}$ on WS$_2$ could induce lattice strain of in-plane W–S phonon mode, featuring a larger number of edge-terminated active sites for promoting HER[42,43]. To reveal the electronic structure of CC@WS$_2$/Pt$_{1,n}$, X-ray photoelectron spectroscopy (XPS) were carried out and shown in Fig. 3c and Supplementary Fig. 6. The survey XPS spectrum of CC@WS$_2$/Pt$_{1,n}$ demonstrated the existence of Pt, W, and S elements on the surface. The high-resolution W 4$f$ spectrum of CC@WS$_2$ showed main peaks located at 32.1 and 34.1 eV were attributed to 4$f_{7/2}$ and 4$f_{5/2}$ of W$^{4+}$ in WS$_2$, respectively. While two other weak peaks at 35.2 and 37.8 eV were assigned to 4$f_{7/2}$ and 4$f_{5/2}$ of W$^{6+}$, resulting from the inevitable surface oxidation when exposing to the air[11,44]. The high-resolution S 2$p$ spectrum of CC@WS$_2$ can be divided into two peaks of 161.6 and 162.8 eV, which were ascribed to S2$^-$ 2$p_{3/2}$ and S2$^-$ 2$p_{1/2}$ in WS$_2$, respectively[45]. After the deposition of Pt$_{1,n}$, the peaks of W 4$f$ and S 2$p$ shifted to lower position, respectively, indicating the strong electronic configuration between Pt$_{1,n}$ and WS$_2$. Therefore, the schottky contact and abundant interface charge between these two components were formed[46]. Compared with standard Pt 4$f$ signal of commercial Pt/C, where the metallic Pt were detected at the location of 71.7 (4$f_{7/2}$) and 75.0 eV (4$f_{5/2}$) with partial surface oxidation, two new peaks positioned at 73.0 (4$f_{7/2}$) and 76.8 eV (4$f_{5/2}$) were attributed to the formation of Pt SAs (Pt$^{\delta+}$, 2 <$\delta$< 4) in CC@WS$_2$/Pt$_{1,n}$, which was consistent with AC-HAADF-STEM analysis[47] (Fig. 3c).

To further reveal the local atomic coordination environments of WS$_2$ and Pt$_{1,n}$ active sites in CC@WS$_2$/Pt$_{1,n}$, X-ray absorption spectroscopy (XAS) was conducted and shown in Fig. 3d–j. First, the X-ray absorption near edge structure (XANES) spectra was performed to explore the oxidation state of the samples. The white line intensity of XANES at the 5$d$ transition metals (e.g., W, Pt) was reliable indicator to their electronic structures (the measured intensity was relevant to the transition of 2$p_{3/2}$ to 5$d_{3/2}$ or 5$d_{5/2}$ because the unoccupied states above Fermi level was crucial to 5$d$ character of W and Pt)[48]. Figure 3d demonstrated that the white line intensity at the W L$_3$-edge for both CC@WS$_2$/Pt$_{1,n}$ and CC@WS$_2$ located between those of WO$_3$ and W foil

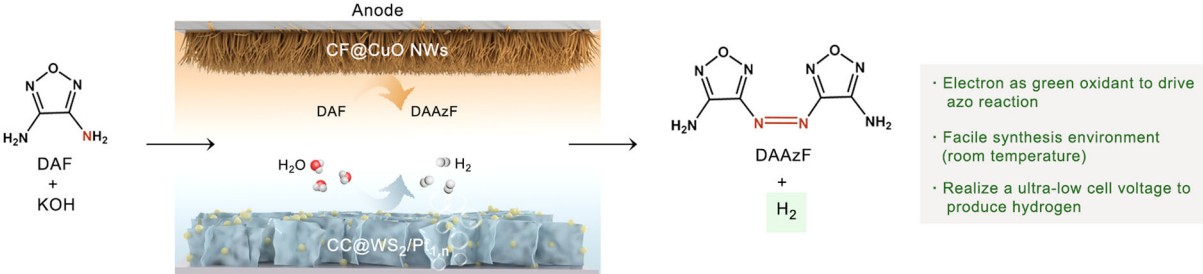

**Fig. 1 | Schematic illustration of traditional and green electrochemical pathways to synthesize DAAzF EMs. a** Traditional pathway to synthesize DAAzF EMs and the corresponding drawbacks. **b** Green electrochemical pathways to synthesize DAAzF EMs and coupled with hydrogen production and corresponding advantages.

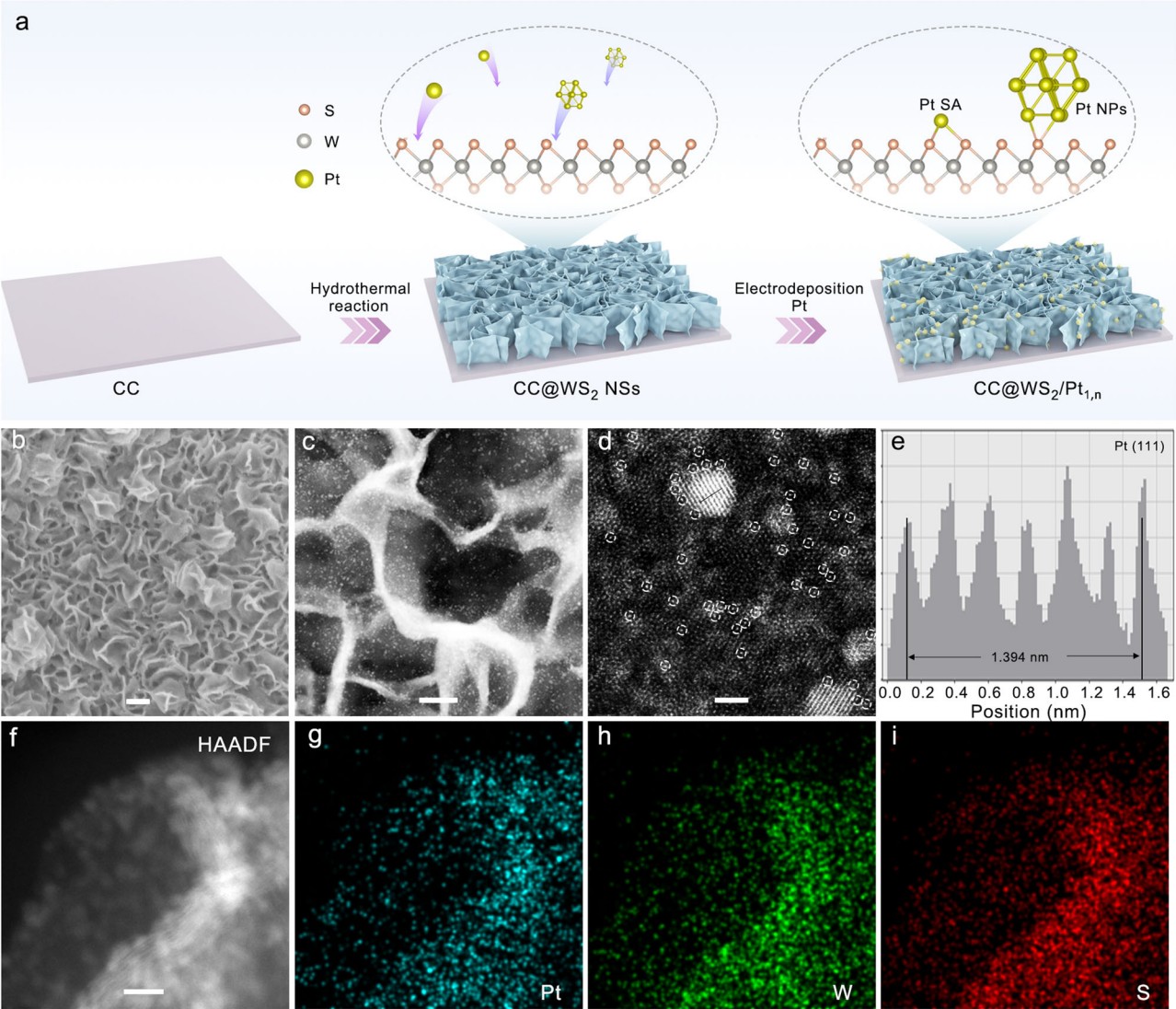

**Fig. 2 | Synthetic procedure and morphological/structural characterizations of CC@WS2/Pt1,n. a** Schematic illustration of synthetic procedure for CC@WS2/Pt1,n. **b** SEM, **c** TEM, **d** AC-HAADF-STEM images of CC@WS2/Pt1,n (scale bar: 200 nm for **b**, 50 nm for **c**, 2 nm for **d**). **e** Corresponding intensity profiles along the line of **d**. **f–i** HAADF-STEM image and elemental mapping of Pt, W, and S of CC@WS2/Pt1,n (scale bar: 10 nm for **f**).

reference samples, suggesting that the valence states of W atoms in CC@WS2/Pt1,n and CC@WS2 were between 0 and +6. Noticeably, the white line intensity of CC@WS2/Pt1,n was lower than that of CC@WS2, indicating the strong electronic configuration between Pt1,n and WS2 in CC@WS2/Pt1,n. This result was in good consonance with XPS analysis. Furthermore, the corresponding Fourier-transformed (FT) $k3$-weighted extended X-ray absorption fine structure (EXAFS) spectra revealed that both CC@WS2/Pt1,n and CC@WS2 showed one prominent peak at 2.0 Å was associated to the W−S path. Another weak peak located at 2.7 Å was attributed to the W−W path. No other W−O interactions at ~1.3 Å could be detected in both CC@WS2/Pt1,n and CC@WS2, demonstrating the pure phase of WS2. In addition, the decrease of intensity at the W−S and W−W shells of CC@WS2/Pt1,n in contrast to CC@WS2 indicated that the deposition of Pt1,n could activate inert basal plane of WS2 and expose more catalytic activated edge sites with dangling bonds for adsorption/desorption of intermediates. The electronic structure of Pt1,n was further explored by the normalized Pt L3-edge XANES spectra. It was noted that the energy of Pt L3-edge was closed to that of W L3-edge, it is difficult to analyze the coordination environment and electronic structure of Pt1,n on the WS2 substrate[49]. Instead, the MoS2 NSs prepared by the same method as

WS2 was the substrate for the deposition of Pt1,n, the structural characterizations of MoS2/Pt1,n were displayed in supplementary Fig. 7. As shown in Fig. 3f, g, the white line-peak intensity of CC@MoS2/Pt1,n with different Pt contents located between Pt foil and PtO2 reference samples, demonstrating the positive charged Pt1,n owing to the strong EMSI leading to the electron transferred from Pt1,n to the substrate, which agreed well with XPS analysis. Moreover, the white line-peak intensity was positive correlation to the ratio of Pt SAs in Pt1,n, where the Pt-0.1 (the Pt4+ content in the electrolyte was 0.1 mL) showed the highest intensity. The white line-peak intensity then decreased and gradually approached to Pt foil with the increase of Pt4+ content to 3.0 mL. This result indicated that the higher concentration of Pt4+ in the electrolyte could lead to the smaller content of Pt SAs and some of Pt SAs were agglomerated to Pt NPs during the electrodeposition of Pt1,n. In order to further reveal the fine coordination environment of Pt1,n, the FT-EXAFS was performed and shown in Fig. 3h. the absence of Pt−Pt contribution at 2.5 Å and the only dominant peak located at 1.9 Å (Pt−S) for Pt-0.1 suggested the atomic dispersed of Pt SAs without Pt NPs in Pt1,n were coordinated with S sites of MoS2[50]. In addition, the attenuation of Pt−S peaks and the rise of Pt−Pt peaks confirmed the more Pt SAs were converted into Pt NPs with the

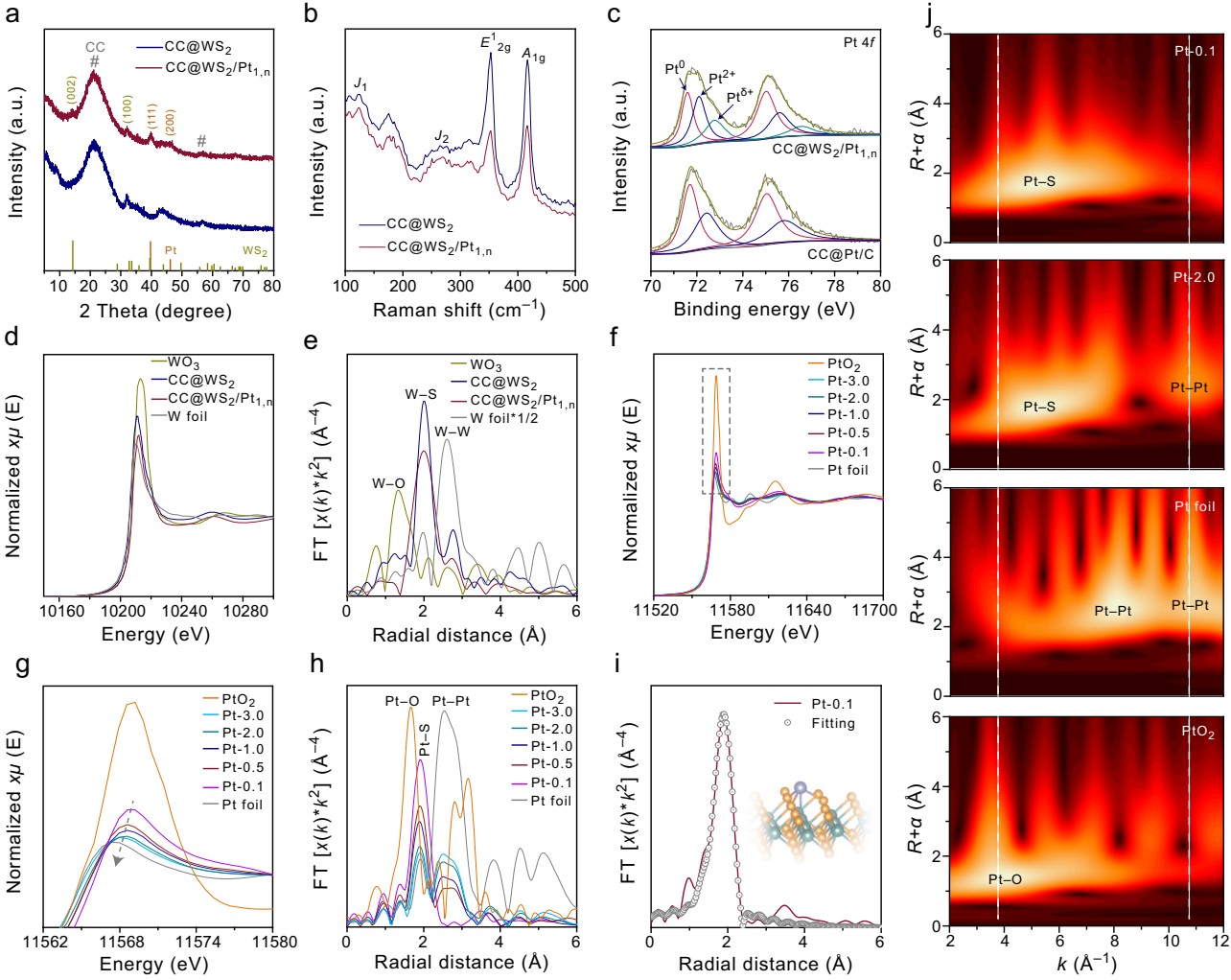

**Fig. 3 | Compositional structure analysis of CC@WS₂/Pt₁,ₙ. a** XRD and **b** Raman spectra of CC@WS₂/Pt₁,ₙ and CC@WS₂. **c** High-resolution XPS spectra of Pt 4*f* signal for CC@WS₂/Pt₁,ₙ and CC@Pt/C. **d** W L₃-edge XANES spectra and **e** FT-EXAFS spectra of CC@WS₂/Pt₁,ₙ and references samples. **f, g** Pt L₃-edge XANES spectra and **h** FT-EXAFS spectra of CC@MoS₂/Pt₁,ₙ with different Pt content and references samples. **i** The fitting of FT *R*-space Pt L₃-edge EXAFS of CC@MoS₂/Pt₁. **j** WT for the $k^3$-weighted EXAFS of CC@MoS₂/Pt₁ and reference samples.

increase of Pt⁴⁺ content during the preparation of Pt₁,ₙ. The fitting results of FT-EXAFS for Pt-0.1 exhibited a coordination number (CN) of 3.1 and 1.1 in the Pt–S and Pt–Mo shells, respectively, suggesting that the Pt SAs coordinated with S at the Mo-top sites of MoS₂ (Fig. 3i and Supplementary Table 1)[36]. The wavelet transform (WT)-EXAFS can directly reflect the *k* and *R* spaces structure information. The WT of FT-EXAFS spectra for Pt-0.1 and Pt-2.0 were shown in Fig. 3g. The WT contour plots of Pt-0.1 displayed only one intensity at ~5.1 Å⁻¹, corresponding to the Pt-light atoms bonding, further confirming the atomic dispersion of Pt in Pt-0.1[51]. Apart from the Pt–S region, Pt–Pt bonds were detected in Pt-2.0 at the *k*-space of 10.8 Å⁻¹, strongly confirming the coexistence of Pt SAs and Pt NPs[49]. Taken together with XAS, AC-HAADF-STEM and XPS analysis, it could be evidenced that Pt₁,ₙ composed of Pt SAs and Pt NPs under a higher Pt⁴⁺ electrolyte concentration (>0.1 mL), and the Pt SAs were well stabilized by the coordination with S atoms on the WS₂ substrate.

**Electrochemical cathodic HER performance**

The catalytic performance of the CC@WS₂/Pt₁,ₙ was investigated by a three-electrode system in 1.0 M KOH. For comparison, the CC@WS₂/Pt₁,ₙ with different Pt contents, CC@WS₂, and commercial 20% Pt/C supported on CC (CC@Pt/C, Supplementary Fig. 8) were also measured under the same condition. The HER performance of CC@WS₂/Pt₁,ₙ

was firstly optimized by adjusting various Pt⁴⁺ contents (0.1–3.0 mL) in the electrolyte during the electrodeposition of Pt₁,ₙ on CC@WS₂ (Fig. 4a). At a current density of 100 mA cm⁻², the over-potential for CC@WS₂/Pt₁,ₙ–0.1, 0.5, 1.0, 2.0, and 3.0 were 191.5, 80.0, 73.0, 60.4, and 66.9 mV, respectively. It was observed that the CC@WS₂/Pt₁,ₙ–0.1 showed insufficient activity, whereas a significant improvement in the HER activity was observed when the Pt⁴⁺ contents higher than 0.1 mL and the 2.0 mL exhibited the best HER performance, indicating the simple Pt SAs sites may not promote the multi-step alkaline HER, the introduction of Pt NPs could induce combined effects of Pt NPs and Pt SAs to provide multiple active sites to accelerate HER activity. The HER polarization curves of different controlled samples were shown in Fig. 4b. The LSV curves of CC@WS₂ showed low activity of 315 mV to deliver current density of 10 mA cm⁻² ($\eta_{10}$), Upon the introduction of Pt₁,ₙ, the drastic HER activity enhancement in the CC@WS₂/Pt₁,ₙ was obtained with a low $\eta_{10}$ of 27.1 mV, which was even lower than those of commercial CC@Pt/C (30 mV) and most of the reported state-of-the-art Pt-based HER catalysts at 10 and 100 mA cm⁻² (Fig. 4c and Supplementary Fig. 9, Tables 2 and 3). In order to further discuss the synergistic effect of Pt NPs and Pt SAs toward HER, the pure Pt NPs supported on the CC@WS₂ (CC@WS₂/Pt NPs) was prepared by chemical reduction method (Supplementary Figs. 10 and 11). The LSV curves of CC@WS₂/Pt NPs was performed and

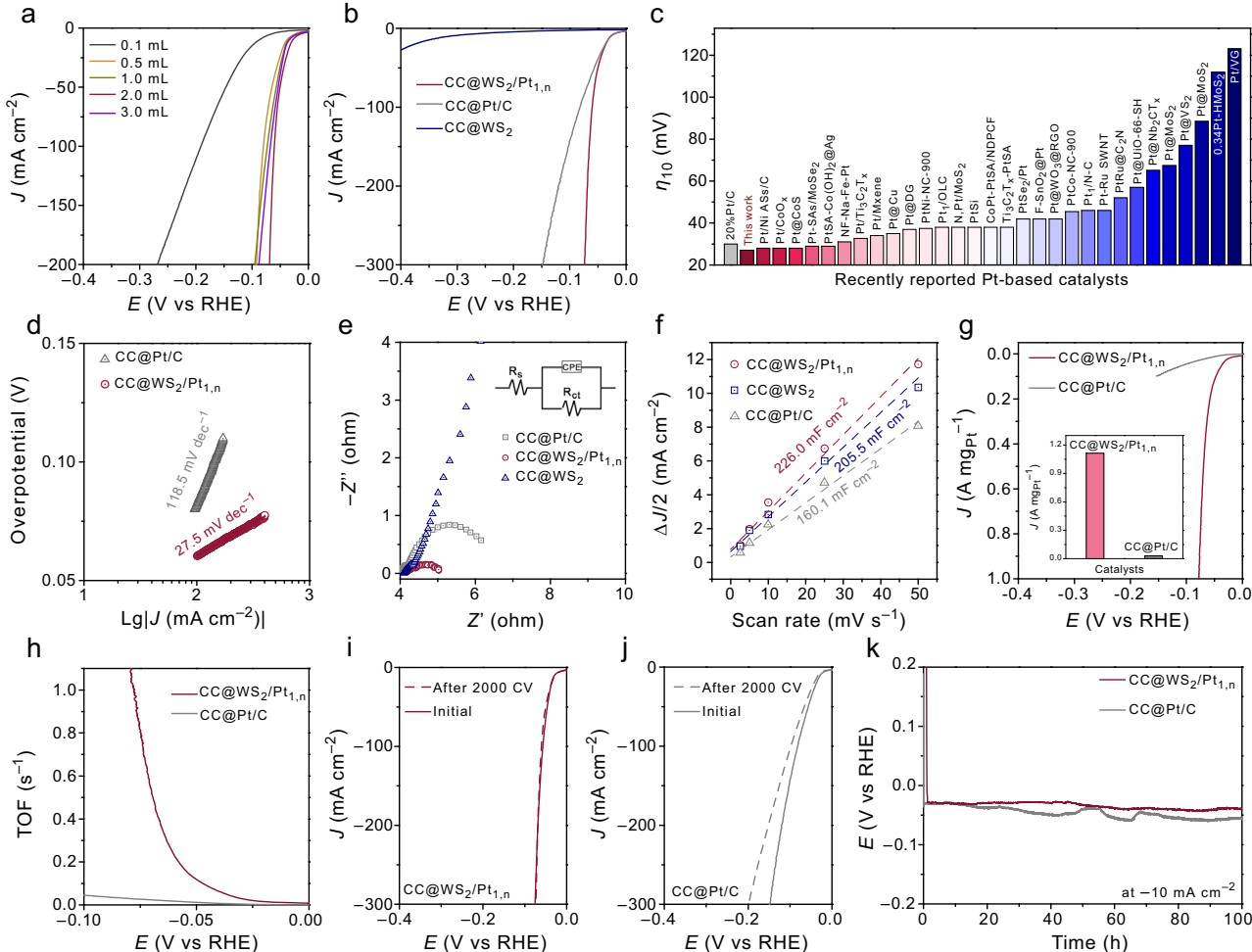

**Fig. 4 | Electrochemical alkaline HER performance. a** LSV curves of CC@WS$_2$/Pt$_{1,n}$ with different Pt$^{4+}$ content in the electrolyte. **b** The LSV curves of optimal CC@WS$_2$/Pt$_{1,n}$ (mass loading: 0.106 mg$_{Pt}$ cm$^{-2}$) and the control samples of CC@WS$_2$ and CC@Pt/C. **c** The comparison of overpotential at the 10 mA cm$^{-2}$ for the prepared CC@WS$_2$/Pt$_{1,n}$ and the reported Pt-based catalysts. **d** Tafel plots of CC@WS$_2$/Pt$_{1,n}$ and CC@Pt/C. **e, f** Nyquist plots and the calculated $C_{dl}$ for the CC@WS$_2$/Pt$_{1,n}$, CC@WS$_2$, and CC@Pt/C, respectively. **g** Mass activities of CC@WS$_2$/Pt$_{1,n}$ and CC@Pt/C on the basis of Pt content. **h** TOFs of CC@WS$_2$/Pt$_{1,n}$ and CC@Pt/C. **i, j** LSV curves before and after 2000 CV tests for CC@WS$_2$/Pt$_{1,n}$ and CC@Pt/C. **k** Chronopotentiometry tests of CC@WS$_2$/Pt$_{1,n}$ and CC@Pt/C. All the tests were measured in 1.0 M KOH solution (pH = 14).

the CC@WS$_2$/Pt$_{1,n}$ was set here for comparison. The $\eta_{10}$ of CC@WS$_2$/Pt NPs (66.0 mV) was significantly higher than that of CC@WS$_2$/Pt$_{1,n}$ (27.1 mV), indicating the combined effect between Pt NPs and Pt SAs for promoting HER (Supplementary Fig. 12). As both Pt NPs and Pt SAs were present in CC@WS$_2$/Pt$_{1,n}$, potassium thiocyanide (KSCN) and ethylenediaminetetraacetic acid disodium (EDTA) were used as poisoning agents to further help differentiate their actual contributions to HER. Note that the EDTA was dominantly coordinated with Pt SAs while SCN$^-$ readily adsorbed onto both Pt SAs and Pt NPs. Thus, EDTA will block and inactivate of Pt SAs sites, and KSCN can deactivate all the Pt sites[34,52]. As shown in Supplementary Fig. 13, with the addition of 10 mM KSCN in the electrolyte, the $\eta_{10}$ value shifted negatively to >180 mV, while the much smaller shift (ca. 11 mV) was observed when 10 mM EDTA was appended to the electrolyte. The different poisoning effect of KSCN and EDTA indicated that both Pt SAs and Pt NPs contributed to the HER, and the latter likely played a dominant role. This result agreed with the theoretical calculations in the following section. To investigate the kinetic behavior and possible reaction pathway for alkaline HER on CC@WS$_2$/Pt$_{1,n}$, the Tafel plots and the corresponding Tafel slopes of each sample were fitted and shown in Fig. 4d. The Tafel slopes of CC@WS$_2$/Pt$_{1,n}$ (27.5 mV dec$^{-1}$) was much lower than that of CC@Pt/C (118.5 mV dec$^{-1}$), suggesting the fast kinetic rate of

Volmer–Tafel pathway with the rate-determining step (RDS) of Tafel that dominated the reaction on CC@WS$_2$/Pt$_{1,n}$[53]. In addition, the electrochemical impedance spectroscopy (EIS) was performed and shown in Fig. 4e. CC@WS$_2$/Pt$_{1,n}$ displayed the smallest charge transfer resistance ($R_{ct}$ = 0.96 Ω) value compared those of CC@Pt/C (2.64 Ω) and CC@WS$_2$ (143.4 Ω), indicating the incorporation of Pt$_{1,n}$ on CC@WS$_2$ can indeed enhance the HER kinetics. The apparent activity of the catalysts were closely associated with the electrochemical active surface area (ECSA), which can be calculated by the electrochemical double-layer capacitance ($C_{dl}$). The $C_{dl}$ was obtained from the CV profiles over a non-Faradic potential range at various scan rates (Supplementary Fig. 14). The calculated $C_{dl}$ of CC@WS$_2$/Pt$_{1,n}$ was 226.0 mF cm$^{-2}$, much higher than those of CC@WS$_2$ (205.5 mF cm$^{-2}$) and CC@Pt/C (160.1 mF cm$^{-2}$). Such a high $C_{dl}$ of CC@WS$_2$/Pt$_{1,n}$ implied an enriched exposed active sites for facilitating the electrochemical HER activity (Fig. 4f).

The intrinsic activity of the catalysts was another crucial criteria for evaluating HER performance. The ECSA-normalized LSV current density for CC@WS$_2$/Pt$_{1,n}$ and CC@WS$_2$ were conducted. The CC@WS$_2$/Pt$_{1,n}$ still showed the highest specific activity with a low overpotential of 70.2 mV than that of commercial CC@Pt/C (105.6 mV) at 10 μA cm$^{-2}$ (Supplementary Fig. 15). The mass activity normalized to

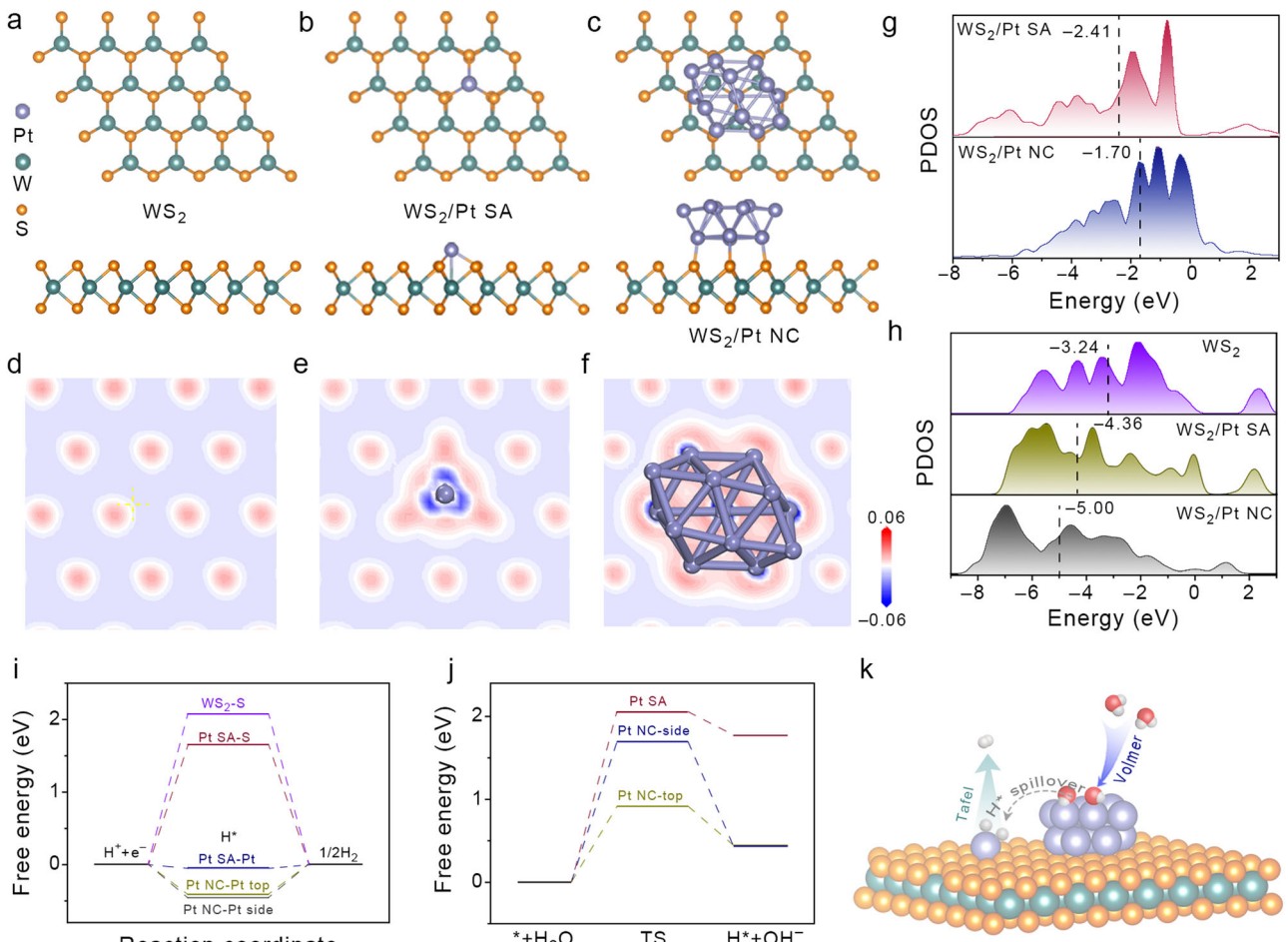

**Fig. 5 | DFT calculations. a–c** The optimized structures of WS₂, WS₂/Pt SA, and WS₂/Pt NC. **d–f** The 2D differential charge density distributions of WS₂, WS₂/Pt SA, and WS₂/Pt NC. Red and blue area represents electron accumulation and depletion, respectively. PDOS of **g** Pt 5*d* for WS₂/Pt SA and WS₂/Pt NC, **h** S 3*p* for WS₂, WS₂/Pt SA, and WS₂/Pt NC. **i** H* adsorption and **j** water adsorption/dissociation free energy on different sites of WS₂, WS₂/Pt SA, and WS₂/Pt NC. **k** Schematic illustration of the alkaline HER mechanism on WS₂/Pt$_{1,n}$.

the loading mass of Pt in CC@WS₂/Pt$_{1,n}$.was conducted to evaluate the economic efficiency of Pt. As displayed in Fig. 4g, at an overpotential of 80 mV, the calculated mass activity of CC@WS₂/Pt$_{1,n}$ was 1.11 A mg$_{Pt}^{-1}$, which were 37 times larger than that of commercial CC@Pt/C (0.03 A mg$_{Pt}^{-1}$). In addition, the turnover frequency (TOF) was calculated to further reveal the intrinsic activity of the samples. Assuming all the Pt sites were all participated in a reaction, the TOF value of CC@WS₂/Pt$_{1,n}$ was 1.16 s$^{-1}$ at an overpotential of 80 mV, which was ~39 times higher than that of CC@Pt/C (0.03 s$^{-1}$). Therefore, the obtained specific activity, mass activity, and TOF calculation demonstrated the significantly enhanced intrinsic activity of CC@WS₂/Pt$_{1,n}$, delivering substantial hydrogen production. For a practical HER electrocatalyst, the stability and long-term durability of CC@WS₂/Pt$_{1,n}$ were further analyzed by the cycling performance and chronopotentiometry (CP) tests. As depicted in Fig. 4i, j, the activity of CC@WS₂/Pt$_{1,n}$ exhibited negligible degradation after 2000 CV, whereas the activity of CC@Pt/C displayed significant deterioration after 2000 CV cycles. Furthermore, the CP measurements were performed and confirmed superior long-term stability of CC@WS₂/Pt$_{1,n}$, where the potential showed ignorable degradation after 100-hour tests under continuous working condition at 10 mA cm$^{-2}$. As a sharp contrast, the apparent activity degradation fluctuation of curve was observed in CC@Pt/C (Fig. 4k). The AC-HAADF-STEM images of CC@WS₂/Pt$_{1,n}$ after the CP test were conducted, the dispersion of Pt NPs and Pt SAs on WS₂ NSs remain nearly unchanged, confirming the stable structure of CC@WS₂/Pt$_{1,n}$ after

long-term stability test, which may due to the strong EMSI effect between Pt$_{1,n}$ and WS₂ substrate to control the dispersion and stabilization of Pt$_{1,n}$ with high surface free energy (Supplementary Fig. 16)[54]. The chronoamperometric (CA) response (at 100 mV) of the CC@WS₂/Pt$_{1,n}$ showed negligible activity decay within 24 hours, the corresponding LSV curves after the CA test was almost the same as that of the initial state, indicating the superior stability of CC@WS₂/Pt$_{1,n}$ electrode (Supplementary Fig. 17). Moreover, the CP test at high current density of –100 mA cm$^{-2}$ was performed and shown in Supplementary Fig. 18. The result indicated that the CC@WS₂/Pt$_{1,n}$ could also maintain long-term stability for at least 100 h, which was due to the in-situ growth of binder-free CC@WS₂/Pt$_{1,n}$ catalysts that can accelerate charge transfer and mass diffusion, promote the exposed active sites, enhance reaction kinetics, and stability[55].

## Theoretical insights of the synergistic HER mechanism

Density functional theory (DFT) calculations were carried out to reveal the underlying mechanism of alkaline HER on WS₂/Pt$_{1,n}$. According to the AC-HAADF-STEM and EXAFS results, the structural models of pristine WS₂, Pt atom located at the top site of W on WS₂ (WS₂/Pt SA), and Pt₁₃ nanoclusters loaded on the WS₂ (WS₂/Pt NC) were built (Fig. 5a–c). Since Pt NPs with an average size of 2.6 nm in our work is nearly impossible for DFT calculations, the Pt₁₃ cluster was adopted for simplicity[50,56–60]. According to the calculated charge density distributions, the increased charge densities could be observed at the WS₂/Pt

SA and WS$_2$/Pt NC interfaces, which implied the strong EMSI and synergistic effect between Pt$_{1,n}$ and WS$_2$ substrate in hybrid. Additionally, the electron transfer occurred at the interfaces, leading to reduced electron density around the Pt atom. To deeply understand the enhancement origin of HER, the projected density of states (PDOS) were conducted. As displayed in Fig. 5g, the $d$-band center ($\varepsilon_d$) is an effective descriptor to directly associate the adsorption properties of surface intermediates and electronic structure of catalysts. The calculated $\varepsilon_d$ of Pt-$d$ orbitals for WS$_2$/Pt SA (−2.41 eV) was more far from to the Fermi level than that of WS$_2$/Pt NC (−1.70 eV), meaning the more moderate interaction between the adsorbed molecules (e.g., H*) and Pt sites of WS$_2$/Pt SA[61]. On the other hand, the enhanced adsorption of intermediates (e.g., OH* in alkaline HER) on WS$_2$/Pt NC intended to promote Volmer reaction during alkaline HER. It was noted that the metal sites may not the only active catalytic centers in some catalysis reactions, the $p$-band center ($\varepsilon_p$) of non-metal sites needed to be revisited and together with $\varepsilon_d$ to synergically regulate electronic structures[62]. In addition, the $\varepsilon_p$ can also be considered as a reliable descriptor for H* adsorption. As shown in Fig. 5h, the $\varepsilon_p$ trend of S-$p$ orbitals states was WS$_2$ > WS$_2$/Pt SA > WS$_2$/Pt NC, indicating the more strengthened adsorption of H* on the S sites of WS$_2$/Pt SA and WS$_2$/Pt NC compared to WS$_2$, which could be in concert with the H* adsorption free energy analysis (see Fig. 5i)[63]. The free energy of H* ($\Delta G_{H*}$) is an important descriptor to evaluate the HER activity, and highly-efficient HER electrocatalysts should possess moderate $\Delta G_{H*}$ near zero. Before the calculation, an ab initio molecular dynamics (AIMD) simulation of WS$_2$/Pt NC was conducted to further prove the stability of the constructed Pt NC on WS$_2$ substrate (Supplementary Fig. 19). As displayed in Fig. 5i and Supplementary Fig. 20, the $\Delta G_{H*}$ of WS$_2$/Pt NC were calculated to −0.416 and −0.463 eV on the top and side sites of Pt NC, respectively, indicating the strengthened H* adsorption on Pt NC may limit the desorption of H* and blocked the active sites for subsequent reaction. The $\Delta G_{H*}$ was reduced to −0.05 eV at the adsorption sites of Pt SA in WS$_2$/Pt SA, indicating the most thermoneutral adsorption of H* (Tafel step). The S sites in both pristine WS$_2$ and WS$_2$/Pt SA showed the $\Delta G_{H*}$ values of 2.069 and 1.648 eV, respectively with more weak adsorption for H* (Supplementary Table 4)[64,65]. It was noted that the |$\Delta G_{H*}$| on Pt SA of WS$_2$/Pt SA was even lower than that of benchmark Pt (111), indicating the well EMSI between Pt SA and WS$_2$ substrate for more thermoneutral adsorption of H*[66]. The energy barrier of water dissociation (Volmer step) is a crucial descriptor to evaluate intrinsic catalytic activity of alkaline HER as the Volmer step is generally considered as RDS of alkaline HER process[67]. As illustrated in Fig. 5j and Supplementary Fig. 21, the water dissociation energy barrier on Pt NC-top sites (0.91 eV) was much smaller related to those of Pt NC-top (1.69 eV) and Pt SA sites (2.05 eV), verifying the accelerated Volmer process on the Pt NC-top sites that dominated the reaction. The above calculation results combined with experimental design demonstrated that the promoted alkaline HER performance could be attributed to the synergistic effect in WS$_2$/Pt$_{1,n}$, is that, the water dissociation kinetic of Volmer step could be promoted on top site of Pt NC, and the obtained H* were spilled and immigrated to adjacent Pt SAs sites of WS$_2$ for hydrogen production (Tafel step) through fast kinetic rate of Volmer−Tafel pathway (Fig. 5k).

## Low-potential anodic DAF oxidation

The low-potential anodic DAF oxidation reaction (DAFOR) is key to our low-energy-consumption hydrogen production system. DAF was used as model substrate for low-potential anodic oxidation reaction. The copper foam supported copper oxide nanowires (CF@CuO NWs) as the catalysts were prepared by a two-step facile method (Fig. 6a). Typically, the copper foam supported copper hydroxide nanowires (CF@Cu(OH)$_2$ NWs) was fabricated by anodic oxidation process, as confirmed by XRD, SEM, and TEM (Supplementary Figs. 22 and 23). The as-formed CF@Cu(OH)$_2$ NWs was then calcined under argon to obtain

CF@CuO NWs. The XRD patterns of CF@CuO NWs in Fig. 6b showed three dominant peaks located at 43.3, 50.5, and 74.1° were attributed to the CF support. The remaining diffraction peaks were well matched with CuO (JCPDS no. 45−0937 and 78−0428). Figure 6c present the SEM images of CF@CuO NWs, the cross-linking nanowires were uniformly and densely grown on CF. The TEM images clearly showed nanowires morphology of CuO with the lattice distances of 0.245 and 0.253 nm, corresponding to the (111) and (002) planes of CuO, respectively. The HAADF-STEM and elemental mapping indicated the homogeneous distribution of Cu and O elements throughout the whole nanowire (Supplementary Fig. 24). The high-resolution XPS spectra of Cu 2$p$ for CF@CuO NWs possessed two main peaks at 933.6 and 953.4 eV can be assigned to 2$p_{3/2}$ and 2$p_{1/2}$ of Cu$^{2+}$ in CuO[68,69]. The shake-up satellites peaks were found at 941.0, 943.6, and 961.9 eV, respectively. High-resolution O 2$p$ spectra consisted of two peaks located at 529.5 and 531.2 eV were attributed to the lattice oxygen (Cu−O−Cu) and adsorbed oxygen in −OH groups, respectively (Supplementary Fig. 25)[69]. Electrocatalytic DAFOR was conducted in a typical three-electrode system using CF@CuO NWs as working electrode, Hg/HgO (1.0 M KOH filler) and Pt wire were reference and counter electrodes. Figure 6d present LSV curves of the CF@CuO NWs in 1.0 M KOH with various concentration of DAF. We found that the DAF oxidation activity was highly dependent on the concentration of DAF. Impressively, the anodic current became gradually obvious when the DAF concentrations changing from 0.02 to 0.20 M. The optimal working potential was 1.23 and 1.44 V $vs.$ RHE at the DAF concentration of 0.20 M to achieve 10 and 100 mA cm$^{-2}$ (Fig. 6e). The low-potential DAFOR showed a promising alternative to traditional OER. To discuss the various catalysts for this reaction, the LSV curves of pristine copper foam (CF) and CF@Cu(OH)$_2$ NWs in 0.20 M DAF + 1.0 M KOH electrolyte were carried out and showed in Fig. 6f. Compared with CF@CuO NWs, the activity attenuation was observed on the CF and CF@Cu(OH)$_2$ NWs catalysts, which may due to the different adsorption capacity of DAF on the catalysts. In view of this, the adsorption energies ($E_{ads}$) of DAF on CF, CF@Cu(OH)$_2$, and CF@CuO were calculated (Supplementary Fig. 26). The $E_{ads}$ of DAF (N sites of furazan ring) bonding with Cu sites of CuO (−1.57 eV) was the lowest compared with those of CuO (N sites of amino bonding with Cu sites of CuO) (−1.34 eV), pristine CF (−1.32 eV), and Cu(OH)$_2$ (−1.48 eV), implying that the Cu sites of CuO as the main adsorption and reaction active site were preferred to bond with N sites of furazan ring in DAF for DAF oxidation[70]. The DAF oxidation mechanism was investigated by the addition of 2,2,6,6-tetramethylpiperidine-1-oxyl (TEMPO) and isopropanol free radical (•OH) scavengers in the electrolyte[71,72]. The polarization curves of DAFOR on CF@CuO NWs in 0.20 M DAF + 1.0 M KOH with and without the addition of TEMPO or isopropanol (Pt film as working electrode to eliminate the poison effect of isopropanol to CuO electrode) were carried out (Supplementary Fig. 27). A significant decrease of current density was detected when TEMPO was added to the electrolyte. In addition, the current intensity was also suppressed after the addition of isopropanol. These results indicated that the DAF oxidation mechanism may involve the oxidative coupling of DAF into DAAzF initiated by the •OH free radical. In presence of applied voltage, the hydroxide anion was converted into •OH free radical, which was acted as oxidant for oxidative coupling of DAF into DAAzF EMs.

## Two-electrode coupling system performance

To establish an energy-efficient system that benefited from the DAFOR, we assembled an two-electrode DAFOR‖HER electrolyser that used CC@WS$_2$/Pt$_{1,n}$ and CF@CuO NWs as cathode and anode catalysts, respectively. The electrolytes contained 1.0 M KOH and 1.0 M KOH + 0.20 M DAF for cathode and anode, respectively (Fig. 7a). Coupling this low-potential DAFOR with a cathodic HER enables a co-production system, with hydrogen production at the cathode side, while the more valuable DAAzF EMs than oxygen was synthesized from DAF starting

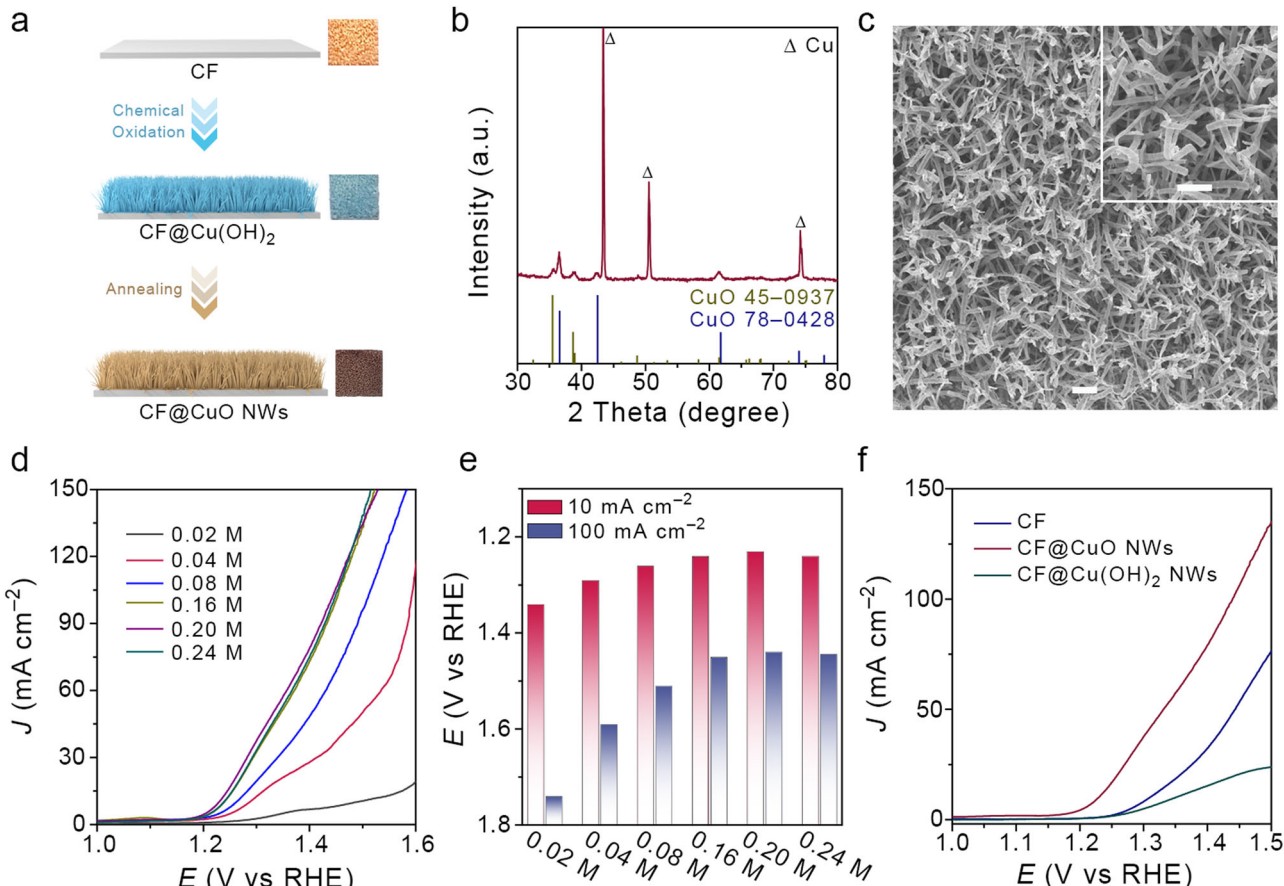

**Fig. 6 | Electrochemical performance of anodic DAF oxidation reaction.**
**a** Schematic illustration of anodic catalysts of CF@CuO NWs. **b** XRD pattern and
**c** SEM images of CF@CuO NWs (scale bar: 2 μm and 1 μm for low and high-
resolution SEM images, respectively). **d** LSV curves of DAF oxidation reaction at the
different concentration of DAF in 1.0 M KOH medium. **e** Corresponding working
potential at the constant current density of 10 and 100 mA cm⁻². **f** LSV curves of DAF
oxidation reaction on different catalysts in 0.20 M DAF + 1.0 M KOH
(pH = 14) media.

material. Figure 7b present the LSV curves of two-electrode electro-
lyser for our novel DAFOR∥HER coupling system and conventional
OWS on CF@CuO NWs∥CC@WS$_2$/Pt$_{1,n}$ catalysts. Remarkably, it only
required the cell voltage of 1.26 V to reach the current density of
10 mA cm⁻², whereas conventional OWS required much higher cell
voltage of 1.67 V to obtain the same current density. This result clearly
indicated energy-efficient advantage of DAFOR than OER. The superior
activity of CF@CuO NWs∥CC@WS$_2$/Pt$_{1,n}$ outperformed the electro-
lyzer employing CF∥CC@Pt/C, CF@Cu(OH)$_2$ NWs∥CC@WS$_2$/Pt$_{1,n}$, and
CF∥CC@WS$_2$/Pt$_{1,n}$ (Fig. 7c). Compared to the recently reported OWS on
noble metal-based catalysts, the DAFOR-assisted OWS proposed in our
work showed significantly reduced cell voltage (Fig. 7d and Supple-
mentary Table 5). To investigate the stability of our coupling system,
the CP test lasting for 40 h was performed. As shown in Fig. 7e, a few
activity attenuation was observed after continuous operation for 40 h.
Furthermore, inspection of the structural and morphological evolu-
tion of anode catalyst after CP test by XRD and SEM showed that the
CuO phase was well maintained. Surprisingly, the CuO nanowires were
aggregated into brush-like morphology (Supplementary Fig. 28). It was
speculated that the activity attenuation was associated with a decrease
in the ECSA because of the morphological evolution of CF@CuO but
did not affect the feasibility of DAFOR to synthesize DAAzF EMs, as
evidenced by the nuclear magnetic resonance (NMR) analysis in the
following section. During the CP test, the hydrogen gas was observed
on the surface of CC@WS$_2$/Pt$_{1,n}$. At the anode side, the color of elec-
trolyte changed from colorless from yellow-orange, preliminarily
supporting the successful synthesis of DAAzF EMs. The obtained

anodic electrolyte was then rotarily evaporated and the product was
characterized by ¹³C NMR (Fig. 7g). The position located at δ = 151.1 and
156.2 ppm were ascribed to −C−NH$_2$ and −C−N=N in furazan ring,
respectively[73]. Additionally, the fourier transform infrared spectro-
scopy (FTIR) of DAAzF EMs exhibited the vibrational peak at 1497 cm⁻¹
was attributed to the $v$(N=N) in DAAzF EMs (Supplementary Fig. 29)[74].
In addition, the Raman spectra was performed to further support the
formation of DAAzF EMs and involved the N−N oxidative coupling of
DAF mechanism during the synthesis of DAAzF (Supplementary
Fig. 30). Furthermore, only one intense exothermic peak located at
327.1 °C was detected in differential scanning calorimetry (DSC)
spectrum (Supplementary Fig. 31). These results confirmed the pure
phase of DAAzF with high energy for the application of energetic
materials. The faradic efficiency (FE) of the cathodic HER and anodic
DAFOR were calculated and shown in Supplementary Fig. 32. The
results confirmed a FE of 98.5% for HER and optimal FE of 94.6% for
DAFOR at 10 mA cm⁻². From the perspective of energy-efficiency, the
electricity input of our coupling system and conventional OWS were
evaluated (Fig. 7h). The conventional OWS required an high electricity
input of 3.98 and 4.76 kWh per m3 of H$_2$ at the current density of 10
and 100 mA cm⁻², and the consumption of electricity significantly
increased at higher current densities, in which the typical electricity
input of ~5 kWh per m3 of H$_2$ was required for conventional OWS at
100 mA cm⁻²[25]. Impressively, because of the low-cell voltage for the
DAFOR∥HER coupling system, the electrolyzer only required an elec-
tricity input of 3.01, 3.35, and 3.70 kWh per m3 of H$_2$ at the current
density of 10, 50, 100 mA cm⁻², respectively, which was much lower

than that of conventional OWS. In view of the low-carbon economy and sustainability, the DAFOR-assisted OWS system provided an energy-efficient, low-cost, and safe strategy for hydrogen production.

## Discussion

In summary, a novel strategy for the concurrent and highly-efficient production of hydrogen and value-added DAAzF EMs was proposed by using CC@WS$_2$/Pt$_{1,n}$ and CF@CuO NWs as cathodic and anodic catalysts, respectively. The low-potential DAFOR in our coupling system was able to realize the low-cell-voltage hydrogen production, offering the current density of 10 mA cm$^{-2}$ at the cell voltage of as low as 1.26 V, which was 410 mV lower than that of conventional OWS. Consequently, this coupling system only required electricity input of 3.01 and 3.70 kWh per m3 of H$_2$ at 10 and 100 mA cm$^{-2}$, respectively, significantly lower than that of conventional OWS. DFT calculations revealed that the Volmer step in cathodic alkaline HER was promoted on the Pt NCs sites of WS$_2$/Pt$_{1,n}$, while the Pt SAs sites exhibited more

thermoneutral H* intermediates adsorption, realizing the synergistic effect of WS$_2$/Pt$_{1,n}$ to promote Volmer–Tafel kinetic in alkaline HER. The anodic DAFOR mechanism involved the oxidative coupling of DAF triggered by •OH free radicals for the synthesis of DAAzF. Apart from the energy-saving hydrogen production, our novel coupling system avoided the traditional hazardous synthetic condition of DAAzF EMs and realized the green and sustainable pathway to electro-synthesize DAAzF EMs. Our work reported a novel alternative reaction of DAFOR to integrate with OWS for energy-saving hydrogen production, and provided a sustainable approach for the production of DAAzF EMs, offering a promising avenue for the development of green electro-synthesis of valuable chemicals coupled hydrogen production systems.

## Methods

### Synthesis of CC@WS$_2$ NSs

CC@WS$_2$ NSs was prepared by one-pot hydrothermal reaction. Typically, 0.90 mmol sodium tungstate dihydrate (Na$_2$WO$_4$·2H$_2$O) and 5.25 mmol

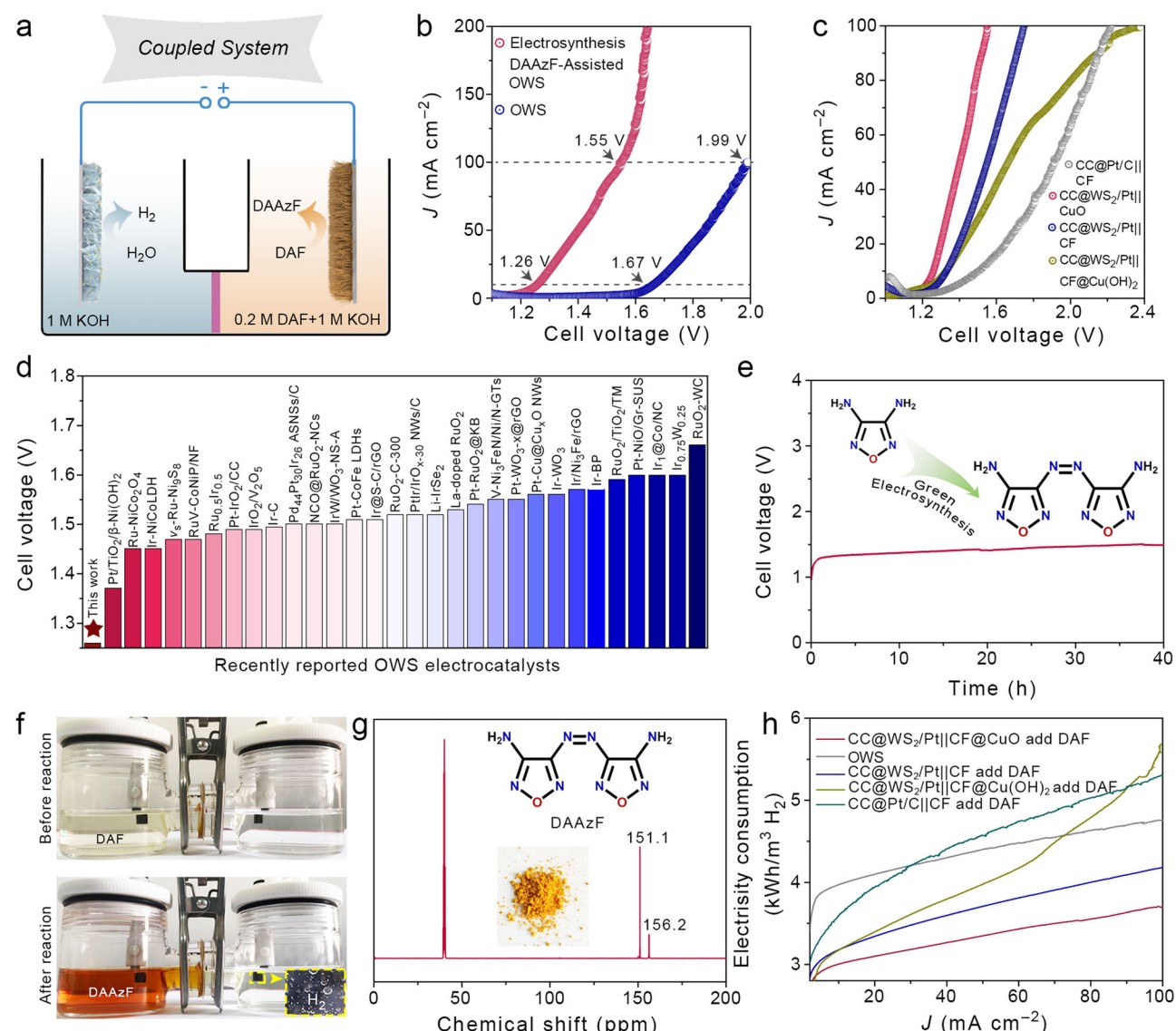

**Fig. 7 | Electrochemical performance of two-electrode coupling system.**
**a** Schematics of the proposed electrocatalytic coupling system. **b** Comparison of LSV curves for conventional OWS and DAF oxidation-coupled OWS system. **c** Comparison of LSV curves of DAF oxidation-coupled OWS on various catalysts. **d** Comparison of the cell voltage required to reach 10 mA cm$^{-2}$ for the coupling system in our work and reported OWS on noble metal-based catalysts. **e** CP curves

of the coupling system recorded at 10 mA cm$^{-2}$ for 40 h. The electrochemical tests were measured in 1.0 M KOH (pH = 14) with or without the addition of 0.2 M DAF. **f** Optical image of our two-electrode coupling system before and after the CP test. **g** $^{13}$C NMR spectrum of the as-synthesized DAAzF EMs (inset: optical image of DAAzF EMs powder). **h** Electricity consumed for hydrogen production in conventional OWS and our coupling system using various catalysts.

thioacetamide ($C_2H_5NS$) were mixed into 15 mL distilled water and stirred for 30 min. Subsequently, 0.15 g oxalic acid ($C_2H_2O_4$) was added to the above aqueous solution and stirred to form a clear solution. A piece of CC (1 cm × 3 cm) was washed with acetone and ethanol several times to clean the surface of CC. After that, the treated CC was immersed into the above solution and transferred to 25 mL Teflon-lined stainless-steel autoclave. The mixture was heated to 200 °C for 24 h. After cooling down to room temperature, the obtained CC@WS$_2$ NSs was washed with distilled water and ethanol, then dried at 60 °C.

## Synthesis of CC@WS$_2$/Pt$_{1,n}$

The as-prepared CC@WS$_2$ NSs was used as substrate to support Pt$_{1,n}$ catalysts. The CC@WS$_2$/Pt$_{1,n}$ was prepared by electrochemical depositions of Pt$_{1,n}$ on CC@WS$_2$ via a cycle voltammetry method. The electrodeposition was carried out in a standard three-electrode system. The self-supporting CC@WS$_2$ was working electrode, a Hg/HgO (1.0 M KOH) was used as reference electrode, and a high-purity graphite rod was used as counter electrode. The 1.0 M KOH with a certain amount of chloroplatinic acid hydrate solution ($H_2Cl_6Pt·xH_2O$, 10 mg mL$^{-1}$) was the electrolyte. The electrodeposition was conducted from −0.6 to −1.5 V vs. Hg/HgO with a sweeping rate of 50 mV s$^{-1}$ for 200 cycles. The content of Pt$_{1,n}$ on CC@WS$_2$ NSs could be controlled by adjusting the volume of $H_2Cl_6Pt$ solution in the electrolyte. For comparison, various volumes of $H_2Cl_6Pt$ (0.1, 0.5, 1.0, 2.0, and 3.0 mL) were set here to investigate the optimal HER performance.

## Synthesis of CC@WS$_2$/Pt NPs

The Pt NPs supported on the CC@WS$_2$ was prepared by one-pot hydrothermal reaction. Ethanol was used as the solvent to dissolve certain amount of $H_2Cl_6Pt$ solution. One piece of CC@WS$_2$ NSs (1.0 cm × 0.5 cm) was loaded at the bottom of the autoclave and heated to 180 °C for 6 h. The final CC@WS$_2$/Pt NPs product was cooled to room temperature and washed with distilled water several times.

## Synthesis of CC@MoS$_2$/Pt$_{1,n}$

The MoS$_2$ nanosheets supported CC (CC@MoS$_2$ NSs) was firstly prepared by one-pot hydrothermal reaction. Typically, the pre-treated (washed by acetone, ethanol several times) CC substrate (1 × 3 cm$^2$) was immersed in the 15 mL aqueous solution containing 1.0 mmol $Na_2MoO_4·2H_2O$ and 4.0 mmol thiourea. The mixture was then transferred to autoclave and heated at 200 °C for 24 h. After cooling down to the room temperature, the obtained CC@MoS$_2$ NSs was used as the support to deposit Pt$_{1,n}$. The CC@MoS$_2$/Pt$_{1,n}$ was prepared by the same electrochemical CV methods as CC@WS$_2$/Pt$_{1,n}$ except for the change of WS$_2$ to MoS$_2$.

## Synthesis of CC@Pt/C

To prepared the Pt/C ink, the commercial 20% Pt/C powder (4 mg) was dispersed in 2.0 mL ethanol containing 50 µL Nafion solution (DuPont, D520, 5%). The suspension was sonicated for 30 min and then dripped on the CC (1.0 cm × 0.5 cm) and dried at 60 °C overnight.

## Synthesis of CF@Cu(OH)$_2$ NWs and CF@CuO NWs

The CF@Cu(OH)$_2$ NWs was prepared by a in-situ oxidative etching method at room temperature. Before the reaction, the CF substrate (1 cm × 0.5 cm) was pre-treated by dilute hydrochloric acid and acetone to remove the surface oxides and organic matters. Then the cleaned CF was immersed in the mixed aqueous solution containing 0.13 M ammonium persulfate (($NH_4)_2S_2O_8$) and 2.67 M NaOH and soaked for 30 min. The obtained CF@Cu(OH)$_2$ NWs was rinsed with ethanol and distilled water for several times and dried at 60 °C for 1 h. The as-prepared CF@Cu(OH)$_2$ NWs was then calcined in muffle furnace at 300 °C for 1 h under air atmosphere to harvest CF@CuO NWs.

## Synthesis of DAF substrate material

The DAF substrate material was synthesized by the two steps from commercial glyoxal as the previous literature[75]. Typically, diaminoglyoxime was firstly synthesized by immersing glyoxal in hydroxylamine hydrochloride (4 equivalents) and sodium hydroxide (4 equivalents) in water at 90 °C. Then, the obtained diaminoglyoxime (23.6 g, 0.2 mol) was dissolved in aqueous potassium hydroxide (2 M, 80 mL) and transferred to a stainless-steel reactor to heat at 170 °C for 2 h. After that, the stainless-steel reactor was cooled at 0-5 °C for 2 h, the obtained white crystals were treated by suction filtrating and vacuum drying to yield DAF substrate material.

## Material characterizations

XRD patterns were obtained from an X-ray diffractometer (Bruker D8 Advance). The morphology of the samples were obtained using Hitachi, SU8010. TEM images and elemental mappings were conducted by FEI Talos F200X. Atomic-resolution HAADF-STEM images of Pt$_{1,n}$ were obtained using FEI Themis Z. The electronic structure of the samples were investigated by XPS (Thermo Scientific Nexsa). Raman spectroscopies of the samples were performed at Thermo Fisher DXR2 Xi acquired by a green laser of 532 cm$^{-1}$. The DSC was conducted on a Q2000/Q600-TA instruments under N$_2$ atmosphere over the heating rate of 10 °C min$^{-1}$. The $^{13}C$ NMR spectrum was carried out on a Bruker Advance 400 in D$_2$O at 25 °C. FTIR spectra was acquired on a Bruker Vertex 70 instrument. The content of Pt$_{1,n}$ in the sample was determined by the ICP-MS (Agilent 7800). Raman spectra were obtained on a Thermo Fisher-Dxr 2Xi. The X-ray absorption spectra were conducted at the beamline 1W1B of the Beijing Synchrotron Radiation Facility (BSRF, operated at typical energy of the storage ring of 2.5 GeV). The X-ray absorption data was collected in the mode of fluorescence for Pt and W L-edges. The least-squares EXAFS fitting was performed using the ARTEMIS module of Demeter package.

## Electrochemical characterizations

The cathodic HER performance of the samples were conducted in 1.0 M KOH electrolyte on a CHI 760E electrochemistry workstation. The as-prepared self-supporting samples were used as working electrode. The high-purity graphite rod and Hg/HgO (1.0 M KOH filler) were counter and reference electrodes, respectively. All potentials were referenced to RHE by Eq. (1):

$$E_{RHE} = E_{Hg/HgO} + 0.098V + 0.0591 × pH \quad (1)$$

The LSV curves were carried out with a scan rate of 2 mV s$^{-1}$. CV tests were performed at the non-faradic potential range of 0.124-0.224 V vs. RHE with the scan rates of 2.5, 5, 10, 20, and 50 mV s$^{-1}$. EIS spectra were obtained in the frequency range of 0.1-100,000 Hz at the overpotential of 200 mV. The stability tests were performed by repeating the CV scan from 0.324-−0.576 V vs. RHE for 2000 cycles. The ECSA of the samples were calculated based on the Eq. (2):

$$ECSA = \frac{C_{dl}}{C_s} \quad (2)$$

Where the $C_{dl}$ is double-layer capacitance, the $C_s$ is the a specific capacitance (0.04 mF cm$^{-2}$)[76]. CP test was performed at a constant current density of −10 mA cm$^{-2}$. Mass activities of the Pt$_{1,n}$ in the samples were calculated by the Eq. (3):

$$Mass\ activity = \frac{J}{m_{Pt}} \quad (3)$$

Where the $J$ is measured current density of Pt, the $m_{Pt}$ is content of Pt in the samples. The TOF of the samples were calculated based on

the Eq. (4):

$$TOF = \frac{I}{2Fn} \qquad (4)$$

Where $I$ is measured current of polarization curve. The $F$ is the Faraday constant (96485 C mol$^{-1}$), the factor 2 is the number of transferred electrons after the formation of one hydrogen molecule. The n is the number of active sites[77]. The anodic DAFOR performance was conducted in a three-electrode system. The as-prepared CF, CF@Cu(OH)$_2$ NWs, and CF@CuO NWs self-supporting catalysts were used as the working electrodes. The Pt wire and Hg/HgO (1.0 M KOH filler) were counter and reference electrodes, respectively. The electrolyte contained 1.0 M KOH with various volume of Pt$^{4+}$ solution (10 mg mL$^{-1}$, 0.1, 0.5, 1.0, 2.0, and 3.0 mL). The LSV curves were carried out with a scan rate of 2 mV s$^{-1}$ over a potential range of 0.724 ~ 1.724 V vs. RHE. The two-electrode coupling system performance were performed in a H-type electrolyzer. The as-prepared Cu based catalysts were used as anode electrodes. The CC@WS$_2$/Pt$_{1,n}$ and CC@Pt/C were cathode electrodes. The electrolyte for anode and cathode were 1.0 M KOH + 0.2 M DAF and 1.0 M KOH, respectively. A piece of anion-exchange membrane (Fumasep FAA-3-50) was used to divide anode and cathode. The LSV curves of the samples were conducted with a scan rate of 2 mV s$^{-1}$. The FE of the HER and DAFOR were calculated based on the following Eq. (5):

$$FE(\%) = \frac{n_{experimental\ produced}}{n_{theoretical produced}} \times 100\% \qquad (5)$$

Where $n$ is the mole value of substrate, the theoretical produced amount of the hydrogen and DAAzF were calculated by the Eq. (6):

$$n_{theoretical peoduced} = \frac{Q}{nF} \qquad (6)$$

Where $Q$ is the charge passed during the reaction, $F$ is Faradic constant (96485 C mol$^{-1}$), $n$ is the number of electron transferred after the formation of each product molecule ($n = 2$ and 4 for HER and DAFOR, respectively). The electricity input ($W$) of the coupling system per m$^3$ of H$_2$ produced (kWh per m$^3$ H$_2$) was calculated by the Eq. (7):

$$W = \frac{n \times F \times U \times 1000}{3600 \times V_m} \qquad (7)$$

Where the $n$ is the number of electrons transferred for H$_2$ production ($n = 2$), $U$ is the applied cell voltage in the two-electrode cell, $V_m$ is the molar volume of H$_2$ at normal temperature and pressure (22.4 mol L$^{-1}$), $F$ is Faraday constant (96485 C mol$^{-1}$).

#### Computational method

The DFT calculations were partially performed via Dmol[3] package of Materials Studio program[78]. The generalized gradient approximation of Perdew−Burke−Ernzerhof exchange correlation function was used. A periodic supercell (4 × 4) and with a vacuum region of 15 Å were added to avoid the interaction between WS$_2$ and its periodic image. An energy convergence criterion of 10$^{-5}$ Hartree was set for geometrical optimization. For the construction of cathodic WS$_2$/Pt SA and WS$_2$/Pt NC catalysts, one Pt atom or Pt$_{13}$ clusters was supported on the surface of WS$_2$ (002) plane. To evaluate the adsorption capacity of DAF on anodic catalysts, one DAF molecule was loaded on the pristine Cu (100), CuO (111), and Cu(OH)$_2$ (002) planes, respectively. 2 × 2 × 1 Monkhorst−Pack $k$-points were sampled for the geometrical optimization.

The transition state for water dissociation step (Volmer) on the Pt sites was determined by the climbing image nudged elastic band method[79]. The adsorption Gibbs free energy of hydrogen atom ($\Delta G_{H^*}$) was calculated by Vienna ab initio simulation package (VASP) code[80] via the Eq. (8):

$$\Delta G_{H^*} = E_{surf + H} - E_{surf} - \frac{1}{2} E_{H_2} + \Delta E_{ZPE} - T \Delta S \qquad (8)$$

Where the $E_{surf + H}$ and $E_{surf}$ represent the total energies before and after the adsorption of H atom, $E_{H_2}$ was the total energy of hydrogen gas molecule. $\Delta E_{ZPE}$, $T$, and $\Delta S$ are the difference in the zero-point energy, temperature, and the difference in entropy, respectively. The effect of implicit solvation was considered by using VASPsol software for simulating the H$_2$O solvent environment and the dielectric constant was set to $\varepsilon = 80$ for water[81].

## Data availability

All data are available from the corresponding authors upon reasonable request. Source data are provided with this paper.

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

## Acknowledgements

This work has received funding from the National Natural Science Foundation of China (52101230 (J.L.), 22102007 (M.X.)), the Shaanxi Key Science and Technology Innovation Team Project (2022TD-33 (Ha.M.)), the Shaanxi Innovation Capability Support Program-Young Science and Technology Star Project (2023KJXX-036 (J.L.)), the Shaanxi Key Laboratory of Special Fuel Chemistry and Material (SPCF-SKL-2021-0004 (J.L.)), the Young Elite Scientists Sponsorship Program by Xi'an Association for Science and Technology (959202313084 (J.L.)), and the Fundamental Research Funds for the Central Universities (buctrc202112 (M.X.)). The authors thank professor Bingbing Suo, professor Haiyan Zhu, and Mr. Jiezhen Xia for the help of VASP calculations, analysis, and discussions. The authors are thankful for the support of the BSRF (Beijing Synchrotron Radiation Facility) during the XAFS measurements at the beamline 1W1B, 4B7A, 4B7B, 4B9A, 4B9B.

## Author contributions

J.L. and Ha.M. conceived the idea. J.L., Y.M., Co.Z., and N.L. synthesized and characterized the materials, and tests the electrochemical performances. J.L. and Ch.Z. performed the theoretical calculations. J.L. and Hu.M. performed the TEM characterizations. J.L. and Z.G. performed the 13C NMR and FTIR characterizations. J.L. and M.X. performed the XAS spectra. J.L. drew the schematic illustrations, wrote the manuscript, and arranged the text layout. Ha.M. and J.Q. revised the manuscript and supervised the work. All the authors discussed the results and review the manuscript.

## Competing interests

The authors declare no competing interests.
