## [Peer review file · Nature Communications]

REVIEWER COMMENTS

Reviewer #1 (Remarks to the Author):

The manuscript describes a novel green electrooxidation coupled system to synthesize 3,3'-diamino-4,4'-azofurazan value-added energetic material starting with 3,4-diaminofurazan and a simultaneous H₂ production on a WS₂/Pt_{1,n} catalyst. The experimental work is well carried out and adequately supported by theoretical calculations. The cathodic catalysts material is characterized enough using comprehensive techniques such as SEM, TEM, AC HAADF-STEM, XRD, XAS, etc. The elaborate WS₂/Pt_{1,n} catalyst shows a low overpotential of 27.1 mV to deliver -10 mA cm⁻² and long-term stability of 100 hours for alkaline HER. The superior alkaline HER performance of WS₂/Pt_{1,n} is attributed to the synergistic effect among the WS₂, Pt SAs and Pt NPs, which is identified from experimental design and theoretical studies. Finally, the integrated novel coupling system delivers ultra-low cell voltage compared with conventional OWS system, revealing a promising energy-saving H₂ production application.

The manuscript thus describes an interesting application of green electrochemistry strategy of producing energetic organic compounds to avoid traditional hazardous synthetic condition and assists H₂ production. This is a typical work about advanced HER catalytic material design and low-energy-consumption H₂ production model. Some minor deficiencies in the work need to be addressed before publication in Nature Communications. The detailed comments are listed below.

- (1) Why selected WS₂ as the substrate rather than other 2D chalcogenides or carbon-based materials to load Pt SAs and Pt NPs catalysts? It is necessary to state the reason in Introduction section.
- (2) The authors selected MoS₂ to replace WS₂ as substrate to deposit Pt for the discussion of XAS characterization due to the energy overlap between W- and Pt L edges. The preparation method and some structural characterizations for MoS₂/Pt such as XRD, SEM, TEM etc. are suggested to be performed.
- (3) EIS Nyquist plots are suggested to be fitted with an equivalent circuit model.
- (4) The authors have provided the stability analysis of the catalyst by chronopotentiometry analysis at 10 mA/cm² current density. Please conduct the chronoamperometric analysis at 100 mV (at least for 24 hrs) to evaluate the stability of the material. Also, provide the HER LSV taken before and after the durability analysis and compare the change in overpotential.
- (5) The calculation details of electricity input of the coupling system should be provided.
- (6) There are some technical errors in the text that should be corrected, eg. Page 4: "synthetic organic electrochemistry" should be "synthetically organic electrochemistry"; The font size of Fig. 1 and Fig. 5i-k were too small to identify; The tenses in the text should be consistent.

Reviewer #2 (Remarks to the Author):

The authors reported various DFT results and discussion, but the following issues should be addressed.

1. In Fig. 5c, provide reasons why the structure of Pt₁₃ is like that. Is there evidence that it is the most stable structure?
2. The G_H values are -2.7 on Pt NC, which seems too low. Can you suggest based on other literature that this value is reasonable?
3. The G_H value for pristine WS₂ is too low. Can you suggest based on other literature that this value is reasonable?

Overall, since this is not the first calculation, it is necessary to compare it with existing literature values. There are already many calculations for WS₂, so please compare them by creating a table. And provide a rationale for using the structure. Also specify information about phases like 2H, 1T, 1T'.

Reviewer #3 (Remarks to the Author):

In this manuscript, Li et al. reports hydrogen production coupled with green electrosynthesis of 3,3'-diamino-4,4'-azofurazan, which is enabled by a single atom Pt catalyst supported on WS₂. Although the idea decoupling HER and OER is not new, the part of herein reported electrosynthesis of furazan is interesting. However, the work in this manuscript is not well presented, and yet may be too preliminary to be publishable. A few major issues have to be addressed.

1. The authors threw in too many contents in this single manuscript. For instance, by the title and introduction, I was expecting an exciting story about the azo reaction. However, the main part of the manuscript is actually talking about HER on a single-atom catalyst. Engineering wise, the authors may have misunderstanding in the hydrogen industry, the whole paper is describing trivial aspects (see details in comment 2 below). Science wise, the single atom catalyst part does not really bring in new science, while the expected azo reaction part is completely missing.

2. The authors attempted to introduce a valuable technology for hydrogen production coupled with electrosynthesis. The whole work is talking about reactions at 10 mA cm⁻² and the overpotential at this current density (η_{10}). For industry, such low current density is not even close to any useful application. PEM and AEM water electrolyzers work at 1 A cm⁻² and above, while the alkaline water electrolyzers work at 100 mA cm⁻² minimum. The η_{10} is meant for comparing model catalysts such as single crystals. For high surface area catalysts, comparing η_{10} does not provide any useful information. Additionally, the energy consumption at such low current density means nothing (Line 25, 26, and 504). The benchmark is not reasonable, as the state-of-the-art water electrolyzer offers at least 1 A cm⁻² at 1.7 V, not 10 mA cm⁻².

3. HER part, the experiment was not correctly done. The Tafel slope of Pt/C should be 120 mV dec⁻¹. Therefore, all the analysis based on the problematic experiment is certainly problematic. Pt/C should be very stable for HER, and thus the results shown in Fig. 4j is questionable. Again, η_{10} is only useful for well-defined catalysts. Therefore, based on presented results, I cannot come out the conclusion that CC@WS₂/Pt_{1,n} is an advanced HER electrocatalyst.

4. Line 444, I am not sure about that the control experiment for the radical mechanism. CuO is a poor catalyst for isopropanol electrooxidation, and therefore isopropanol may be a poison for CuO which may also cause the reduced current.

5. Again, reactions (HER, DAFOR, and coupled one) at 10 mA cm⁻² is not of interest to industry. The authors have to present results at high current density to showcase the practical value of their system. Additionally, increasing energy efficiency by reducing the operation voltage (or, current density) is nonsense.

6. Fig. 7d is useless but misleading. If HER coupled with alcohol oxidation, the overall voltage can be reduced to a value way below 1 V. The point of decoupled HER/OER is not at how low voltage you can produce hydrogen, it is how much value you can add by the coupled reaction.

Overall, the work has potential, and the story could be much more exciting by focusing on the azo reaction.

Dear Editor:

Much thanks for all the comments from the reviewers, which helps us to improve our manuscript a lot. We listed the answers in the following text.

Response to Reviewer #1:

Comments:

The manuscript describes a novel green electrooxidation coupled system to synthesize 3,3'-diamino-4,4'-azofurazan value-added energetic material starting with 3,4-diaminofurazan and a simultaneous H₂ production on a WS₂/Pt_{1,n} catalyst. The experimental work is well carried out and adequately supported by theoretical calculations. The cathodic catalysts material is characterized enough using comprehensive techniques such as SEM, TEM, AC HAADF-STEM, XRD, XAS, etc. The elaborate WS₂/Pt_{1,n} catalyst shows a low overpotential of 27.1 mV to deliver -10 mA cm⁻² and long-term stability of 100 hours for alkaline HER. The superior alkaline HER performance of WS₂/Pt_{1,n} is attributed to the synergistic effect among the WS₂, Pt SAs and Pt NPs, which is identified from experimental design and theoretical studies. Finally, the integrated novel coupling system delivers ultra-low cell voltage compared with conventional OWS system, revealing a promising energy-saving H₂ production application.

The manuscript thus describes an interesting application of green electrochemistry strategy of producing energetic organic compounds to avoid traditional hazardous synthetic condition and assists H₂ production. This is a typical work about advanced HER catalytic material design and low-energy consumption H₂ production model. Some minor deficiencies in the work need to be addressed before publication in Nature Communications. The detailed comments are listed below.

Response:

We express our sincere gratitude to the reviewer for your careful review on our manuscript and the constructive comments, which really help to improve the quality of our manuscript. Following the reviewer's suggestions, the required characterizations of the catalysts have been supplied. We hope that our changes will satisfy the reviewer.

Comment #1

Why selected WS₂ as the substrate rather than other 2D chalcogenides or carbon-based materials to load Pt SAs and Pt NPs catalysts? It is necessary to state the reason in Introduction section.

Response:

In industrial heterogeneous catalysis, the metal nanoparticles supported on the substrate could regulate the electronic structure of active sites through electronic metal-support interaction (EMSI). Compared with carbon-based substrates, the transition metal dichalcogenides (TMDs) supported metal

single atoms/nanoparticles could adjust the electronic structure through both anchoring atom and the adjacent transition metal atoms with higher atomic number, which provided more flexible local coordination environment to regulate the catalytic activity (*Adv. Mater.* 2020, 32, 2003300). The core anchoring chalcogen (S) and the adjacent transition metal (W) can synergistically regulate the *d*-orbital state electronic structure of metal single atoms/nanoparticles through EMSI. As a result, the adsorption energy of intermediates on active sites could be optimized and thus has influence on the catalytic activities (*Nat. Commun.* 2021, 12, 3021). As for one of the important candidates in TMDs, the metallic WS₂ achieved excellent catalytic activity for HER compared with other TMDs because the basal plane of metallic WS₂ was also catalytically active than natural MoS₂ (*Energy Environ. Sci.*, 2014, 7, 2608; *J. Phys. Chem. C*, 2020, 124, 789–798). The periodic density functional theory calculations also indicated that the catalytic activity of WS₂ was predicted to be comparable to or even better than MoS₂ (*Phys. Chem. Chem. Phys.* 2014, 16, 13156–13164). Therefore, the WS₂ could be both a better substrate for Pt_{1,n} deposition and potential catalyst for HER. This section was added to the Introduction part in Page 5 in the revised manuscript.

Comment #2

The authors selected MoS₂ to replace WS₂ as substrate to deposit Pt for the discussion of XAS characterization due to the energy overlap between W- and Pt L edges. The preparation method and some structural characterizations for MoS₂/Pt such as XRD, SEM, TEM etc. are suggested to be performed.

Response:

Thanks for your suggestion. We have supplied the preparation method and conducted structure characterizations and made some revisions.

The MoS₂ nanosheets supported CC (CC@MoS₂ NSs) was firstly prepared by one-pot hydrothermal reaction. Typically, the pre-treated (washed by acetone, ethanol several times) CC substrate (1×3 cm²) was immersed in the 15 mL aqueous solution containing 1.0 mmol Na₂MoO₄·2H₂O and 4.0 mmol thiourea. The mixture was then transferred to an autoclave and heated at 200 °C for 24 h. After cooling down to the room temperature, the obtained CC@MoS₂ NSs was used as the support to deposit Pt_{1,n}. The CC@MoS₂/Pt_{1,n} was prepared by the same electrochemical CV methods as CC@WS₂/Pt_{1,n} except for the change of WS₂ to MoS₂. This section was added to the Methods part.

The morphological and structural characterizations of CC@MoS₂ and CC@MoS₂/Pt_{1,n} were performed and shown in Fig. R1. XRD pattern revealed the structure of MoS₂ substrate still stable after the deposition of Pt_{1,n} (Fig. R1a). In addition to the diffraction peaks for MoS₂, a distinct peak located at ~40° was attributed to the Pt (111) phase of Pt_{1,n}, indicating the successful loading of Pt_{1,n} on CC@MoS₂ NSs. SEM image showed the cross-linking of MoS₂ NSs with smooth surface. After the deposition of Pt_{1,n}, the rough surface of MoS₂ NSs was observed without the large Pt nanoparticles, suggesting that the ultra-small size of Pt_{1,n} were uniformly dispersed on the MoS₂ NSs (Fig. R1b, c). In addition, the TEM and HRTEM images showed that the ultra-small Pt_{1,n} (direction of arrow) were loaded on the corrugated MoS₂ NSs (Fig. R1d). The corresponding HAADF-STEM image and elemental mappings showed homogeneously dispersed of Mo, S, and Pt on the CC@MoS₂/Pt_{1,n} (Fig. R1e~h). The above discussion supported the successful preparation of CC@MoS₂/Pt_{1,n}. This section

was added to Supplementary Fig. S7 in the revised supporting information.

Fig. R1. **a** XRD pattern of CC@MoS₂/Pt_{1,n}. **b, c** SEM images of CC@MoS₂ and CC@MoS₂/Pt_{1,n} (scale bar: 200 nm for **b** and **c**). **d** TEM images of CC@MoS₂/Pt_{1,n} (scale bar: 20 nm). **e-h** HAADF-STEM image and elemental mappings of Mo, S, and Pt on CC@MoS₂/Pt_{1,n} (scale bar: 50 nm).

Comment #3

EIS Nyquist plots are suggested to be fitted with an equivalent circuit model.

Response:

We made the fitting as you suggested. The EIS Nyquist plots of CC@WS₂/Pt_{1,n}, CC@Pt/C, and CC@WS₂ were fitted with the equivalent circuit models (Fig. R2). The charge-transfer impedance (R_{ct}) of the HER can be obtained by fitting the curves with the equivalent circuit. The CC@WS₂/Pt_{1,n} showed the R_{ct} of 0.96 Ω , which is much lower than those of CC@Pt/C (2.64 Ω) and CC@WS₂ (143.4 Ω), confirming that the CC@WS₂/Pt_{1,n} has a fast electron transfer and HER kinetic at the electrolyte/electrocatalyst interface. This Figure was updated to Fig. 4e and the corresponding discussion was added to Page 16 in the revised manuscript.

Fig. R2. Nyquist plots for CC@WS₂/Pt_{1,n}, CC@WS₂, and CC@Pt/C

Comment #4

The authors have provided the stability analysis of the catalyst by chronopotentiometry analysis at 10 mA/cm² current density. Please conduct the chronoamperometric analysis at 100 mV (at least for

24 hrs) to evaluate the stability of the material. Also, provide the HER LSV taken before and after the durability analysis and compare the change in overpotential.

Response:

Thanks for your suggestion. The chronoamperometric (CA) response (at 100 mV) of the CC@WS₂/Pt_{1,n} showed negligible activity decay within 24 hrs, the corresponding LSV curves after the CA test was almost the same as that of the initial state, indicating the superior stability of CC@WS₂/Pt_{1,n} electrode (Fig. R3). This section was added to supplementary Fig. S17 in the revised supporting information. The corresponding discussion was added to Page 17 in the revised manuscript.

Fig. R3. Long-term stability CA test of CC@WS₂/Pt_{1,n}. (inset: LSV curves before and after the CA test).

Comment #5

The calculation details of electricity input of the coupling system should be provided.

Response:

Yes, we provided the calculation details according to your suggestion. The electricity input (W) per m³ of H₂ produced (kWh per m³ H₂) was calculated by the equation:

$$W = (n \times F \times U \times 1000)/(3600 \times V_m)$$

Where the n is the number of electrons transferred for H₂ production ($n=2$), U is the applied cell voltage in the two-electrode cell, V_m is the molar volume of H₂ at normal temperature and pressure (22.4 mol L⁻¹), F is Faraday constant (96485 C mol⁻¹) (*Nat. Catal.* 2022, 5, 66–73). This section was added to the electrochemical characterizations of the Methods section in the revised manuscript.

Comment #6

There are some technical errors in the text that should be corrected, eg. Page 4: “synthetic organic electrochemistry” should be “synthetically organic electrochemistry”; The font size of Fig. 1 and Fig. 5i-k were too small to identify; The tenses in the text should be consistent.

Response:

Thank you for pointing this out. The noted technical errors were corrected. In addition, the grammatical errors and improper usage of English in the manuscript have been checked out and corrected. Furthermore, in the revised manuscript, we have tried our best to substantially improve the

English. We also asked help from a professor for the language, his help was appreciated in the Acknowledgement.

Response to Reviewer #2:

Comments:

The authors reported various DFT results and discussion, but the following issues should be addressed.

Response:

We express our sincere gratitude to the reviewer for your careful review on our manuscript and the constructive comments, which really help to improve the quality of our manuscript. Following the reviewer's suggestions, the rationale for using the WS₂/Pt_{1,n} structure has been provided and the reasonability of the ΔG_{H^*} has been explained. We hope that our changes will satisfy the reviewer.

Comment #1

In Fig, 5c, provide reasons why the structure of Pt13 is like that. Is there evidence that it is the most stable structure?

Response:

Since Pt NPs with an average size of 2.6 nm in our work is nearly impossible for density functional theory (DFT) calculations, the Pt₁₃ cluster was adopted for simplicity in our present work (*Nat. Commun.* 2022, 13, 6863; *Angew. Chem. Int. Ed.* 2021, 60, 23388–23393). For the initial geometry of Pt₁₃ in our catalysts system, we chose a minimum-energy structure determined by Piotrowski et al, in which the lowest-energy local minimum structures of Pt₁₃ were identified by DFT investigations (*Phys. Rev. B* 2010, 81, 155446). In addition, many recent first-principles calculations based on DFT on Pt₁₃ clusters have been widely studied for hydrogen production (*Nat. Commun.* 2023, 14, 2460; *Adv. Energy Mater.* 2023, 13, 2204213; *ACS Catal.* 2017, 7, 2744–2752;). For a better understanding, the related literatures were cited in Refs. 50, 57–61 in the revised manuscript.

Comment #2

The G_H values are -2.7 on Pt NC, which seems too low. Can you suggest based on other literature that this value is reasonable?

Response:

Thanks for your question. The calculated charge density distributions of Pt NC on WS₂ surface could be come out that the increased charge densities were observed around the Pt NC (Fig. 5f), which implied the strong electronic metal-support interaction (EMSI) that is associated with the orbital rehybridization and charge transfer across the metal–support (Pt NC/WS₂) interface. The electron transfer modulates the *d*-band structure of Pt NC and regulates the adsorption of reaction intermediates (H^{*}). The accumulated charge densities around Pt NC induced too strengthened adsorption of hydrogen atoms to desorb, which resulted in more negative ΔG_{H^*} of Pt NC. This phenomenon is similar to some of the recently reported Pt-group metal-based systems (*Angew. Chem. Int. Ed.* 2023, 62, e202300879 ($\Delta G_{H^*}(\text{Ru}_{13})=-3.6$ eV); *ACS Nano* 2022, 16, 14885–14894 ($\Delta G_{H^*}(\text{Ru}_{\text{hcp}})=-1.57$ eV); *Nat. Commun.* 2020, 11, 1029 ($\Delta G_{H^*}(\text{Pt})=-1.35$ eV); *Nat. Commun.* 2019, 10, 4936 ($\Delta G_{H^*}(\text{PtRu})=-1.0$ eV)). Some of the literatures were cited in Refs. 66–68 in the revised manuscript for a better understanding.

Comment #3

The G_H value for pristine WS₂ is too low. Can you suggest based on other literature that this value is reasonable?

Response:

Thanks for your suggestion. the ΔG_{H^*} of hydrogen atom adsorbed on S sites in pristine WS₂ was calculated to be -0.41 eV, the relatively low ΔG_{H^*} compared with that of S sites in WS₂/Pt SA (-0.29 eV) may due to the inert basal plane part of pristine WS₂. The Pt atoms depositing on WS₂ could keep it as the main active site for H adsorption and activate the basal S atoms simultaneously for more thermoneutral hydrogen adsorption (*Adv. Sci.* 2021, 8, 2002284). This phenomenon regarding to the relatively low ΔG_{H^*} of pristine WS₂ is similar to some of the recently reported WS₂-based electrocatalysts (*Nat. Commun.* 2021, 12, 709 ($\Delta G_{H^*}(2H-WS_2)=-0.78$ eV); *Nat. Commun.* 2021, 12, 5070 ($\Delta G_{H^*}(2H-WS_2)=-0.72$ eV)). The related literatures were cited in Refs. 69,70 in the revised manuscript for a better understanding.

Comments

Overall, since this is not the first calculation, it is necessary to compare it with existing literature values. There are already many calculations for WS₂, so please compare them by creating a table. And provide a rationale for using the structure. Also specify information about phases like 2H, 1T, 1T'.

Reply:

Thanks for your suggestion. The structure of WS₂ in CC@WS₂/Pt_{1,n} was verified by Raman spectra and shown in Fig. 3b. Two strong features located at 352 and 416 cm⁻¹ in CC@WS₂/Pt_{1,n} corresponded to E_{1g} and A_{1g} of 2H phase for WS₂, indicating the 2H-WS₂ dominated the structure (*Adv. Mater.* 2015, 27, 4837–4844). Furthermore, the local coordinate environment of Pt_{1,n} in CC@WS₂/Pt_{1,n} was revealed by Fourier-transformed EXAFS of Pt-L₃ edge (Fig. 3h). The result showed that the Pt SAs and Pt NPs were co-existed in CC@WS₂/Pt_{1,n}, as evidenced by the detected Pt–S and Pt–Pt bonds at 1.9 and 2.5 Å, respectively (*Nat. Commun.* 2022, 13, 6863). In order to further elucidate the local coordinate environment of Pt SAs in CC@WS₂/Pt_{1,n}, the fitting of EXAFS was performed and exhibited a coordination number of 3.1 and 1.1 in the Pt–S and Pt–M shells, respectively, suggesting that the Pt SAs may coordinate with S at the W-top sites of WS₂ (Fig. 3i and Supplementary Table 1) (*Nat. Commun.* 2022, 13, 3063).

Based on the above structural characterizations of CC@WS₂/Pt_{1,n}, the structural models of pristine 2H-WS₂ and Pt SA located the surface of 2H-WS₂ were constructed and fully relaxed for DFT calculations. To construct WS₂/Pt NC structure, Pt₁₃ cluster with lower energy structure was adopted as the alternative in our present work. For a better comparison, the reported calculations for WS₂-based structures were summarized and provided in Table R1. This section was added to the Supplementary Table 4 in the revised supporting information.

Table R1. Comparison of the ΔG_{H^*} on various sites of reported WS₂-based catalysts.

Catalysts	H [*] -adsorption sites	ΔG_{H^*} (eV)	References
WS ₂ /Pt _{1,n}	Pristine 2H-WS ₂ -S site	-0.41	This work
	2H-WS ₂ /Pt SA-S site	-0.29	

V SACs@1T-WS ₂	V SACs@1T-WS ₂ -V site	-0.05	Nat. Commun. 2022, 13, 3063
	V SACs@1T-WS ₂ -S site	-0.21	
	V SACs@2H-WS ₂ -V site	-0.13	
	V SACs@2H-WS ₂ -S site	-0.30	
	Pristine 2H-WS ₂ -S site	-0.78	
WS ₂ MSLs	Pristine 2H-WS ₂ -W site	-0.73	Nat. Commun. 2021, 12, 5070
	Rotated 2H-WS ₂ -W site1	-0.51	
	Rotated 2H-WS ₂ -W site2	-0.24	
	Pristine 2H-WS ₂ -S site	-0.73	
	Rotated 2H-WS ₂ -S site1	-0.25	
	Rotated 2H-WS ₂ -S site2	-0.18	
1T-WS ₂ /a-WO ₃	WO ₃	-2.50	ACS Appl. Mater. Interfaces 2022, 14, 24008–24019
	1T-WS ₂	-0.75	
Pt-SAs/WS ₂	Pt-SAs/2H-WS ₂ -Pt site	-0.06	Nat. Commun. 2021, 12, 3021
noble metal@MoS ₂ /WS ₂	2H-MoS ₂ @Pd@2H-WS ₂	0.04	Appl. Surf. Sci. 2022, 591, 153168
	2H-MoS ₂ @Pt@2H-WS ₂	0.03	
	Pt@2H-MoS ₂ /Pt@2H-WS ₂	0.008	
	Pd@2H-MoS ₂ /Pd@2H-WS ₂	-0.015	
	Pt@2H-MoS ₂ /2H-WS ₂	-0.05	
	Pd@2H-MoS ₂ /2H-WS ₂	-0.084	
WS ₂ @VS ₂	2H-WS ₂ @VS ₂ -W-S site	0.11	Small 2023, 19, 2205881
	2H-WS ₂ @VS ₂ -V-S site	-0.36	
N-WS ₂ /Co ₃ N	N-2H-WS ₂ /Co ₃ N	-0.34	Small 2023, 18, 2203171
	N-2H-WS ₂	-0.64	
2H-WS ₂ /Ru SAs	Sv-2H-WS ₂ -Sv site	-0.74	Adv. Funct. Mater. 2022, 32, 2109439
	Sv-WS ₂ /Ru SAs-Ru site	-0.64	
	Sv-WS ₂ /Ru SAs-S site	-0.21	
P-1T'-WS ₂	0.01P-1T'-WS ₂	-0.35	Int. J. Mol. Sci. 2022, 23, 11727
	0.02P-1T'-WS ₂	-0.37	
	0.03P-1T'-WS ₂	-0.37	
	0.04P-1T'-WS ₂	-0.38	
2H-WS ₂ NSs	C-WS ₂	-1.60	Angew. Chem. Int. Ed. 2021, 60, 21550–21557
	D1-WS ₂ NSs	~-0.75	
	D2-WS ₂ NSs	~-0.75	
	Pristine WS ₂ NSs	~-0.75	
Sv-WS ₂ @DG	Pristine 2H-WS ₂	-1.62	Nano Res. 2022, 15, 677–684
	Sv-2H-WS ₂	-0.82	
	Sv-2H-WS ₂ @DG	-0.41	
1T/2H WS ₂ -2D Ni ₃ S ₄	Pristine 2H-WS ₂ -S site	~-1.70	CCS Chem. 2021, 3, 58–68
	Pristine 1T-WS ₂ -S site	~-0.70	
	2H-WS ₂ -Ni ₃ S ₄ -W-S site	~-0.25	
	1T-WS ₂ -Ni ₃ S ₄ -W-S site	~-0.10	
W/WS ₂	2H-WS ₂ /W ₄ -3-fold site	-0.06	Phys. Chem. Chem. Phys. 2019, 21, 6071–6079
	2H-WS ₂ /W ₄ -top site	-0.24	

	2H-WS ₂ /W ₄ -edge site	-0.88	
	Pristine 1T-WS ₂	-0.40	
	N-1T-WS ₂	-0.32	J. Mater. Chem. A , 2018,6,
N, P-WS ₂	N, P-1T-WS ₂ -edge site	-0.26	19712–19726
	N, P-1T-WS ₂ -P site	-0.63	

Response to Reviewer #3:

Comments:

In this manuscript, Li et al. reports hydrogen production coupled with green electrosynthesis of 3,3'-diamino-4,4'-azofurazan, which is enabled by a single atom Pt catalyst supported on WS₂. Although the idea decoupling HER and OER is not new, the part of herein reported electrosynthesis of furazan is interesting. However, the work in this manuscript is not well presented, and yet may be too preliminary to be publishable. A few major issues have to be addressed.

Response:

We express our sincere gratitude to the reviewer for your careful review on our manuscript and the constructive comments, which really help to improve the quality of our manuscript. Following the reviewer's suggestions, the performance of HER, DAFOR, and coupling system at high current density have been supplied. In addition, the discussion of HER catalysts and DAFOR to electro-synthesize DAAzF EMs were highlighted for a better understanding.

Comment #1

The authors threw in too many contents in this single manuscript. For instance, by the title and introduction, I was expecting an exciting story about the azo reaction. However, the main part of the manuscript is actually talking about HER on a single-atom catalyst. Engineering wise, the authors may have misunderstanding in the hydrogen industry, the whole paper is describing trivial aspects (see details in comment 2 below). Science wise, the single atom catalyst part does not really bring in new science, while the expected azo reaction part is completely missing.

Response:

Thanks for the suggestion about the story brought in our work. The electrooxidation of organic compounds and hydrogen evolution in a hybrid water electrolyzer is a fascinating synthetic strategy that opens a sustainable approach for upgrading organic compounds on the anode. At the same time, renewable H₂ gas is produced more effectively on the cathode. Despite the superior advantages of such a coupling approach and some recent progress that has been made, the coupling technology is just at an early stage and the transformation efficiency is still unsatisfactory. Thus, both the exploration of more alternative reactions to OER and the reasonable design of high-performance catalysts are highly necessary for the development of hybrid water electrolysis (*Angew. Chem. Int. Ed.* 2022, e202209014). In our manuscript, the new electrochemical azo oxidation alternative to OER was introduced to the coupling system. Moreover, we devoted much efforts to the design of highly-efficient HER catalyst that featured co-existence of Pt SAs and Pt NPs on the WS₂ substrate.

In fact, single atoms are not the only configuration of active centers in many single atoms catalysts (SACs), where atomic clusters exist as well, which is usually overlooked (*Nat. Commun.* 2017, 8, 15938). Owing to their extremely high surface free energy, single atoms tend to agglomerate into atomic clusters during synthesis or under working conditions. Therefore, the formation of atomic clusters is more energetically favored and thus more stable under working conditions than single atoms (*ACS Catal.* 2019, 9, 8213–8223). It has been demonstrated that a strong interaction between the deposited metal and the support material plays a vital role in the stabilization of supported catalysts. Despite the controlled deposition process during the preparation of SACs, atomic diffusion and

agglomeration are still possible and probable, if a weak interaction exists between the atoms and the support, thus resulting in the formation of large particles (*Nat. Commun.* 2016, 7, 13638). On the other hand, single atoms do not exhibit their overwhelming advantages in every circumstance such as multi-step catalysis of alkaline HER, because the alkaline HER involves three steps of Volmer step ($\text{H}_2\text{O} + \text{e}^- \rightarrow \text{H}_{\text{ads}} + \text{OH}^-_{(\text{aq})}$) to dissociate water molecules to produce H_{ads} , which are then combined with another H_{ads} to form H_2 (Tafel step, $\text{H}_{\text{ads}} + \text{H}_{\text{ads}} \rightarrow \text{H}_{2(\text{g})}$) or reacted with another water molecule to form H_2 (Heyrovsky step, $\text{H}_{\text{ads}} + \text{H}_2\text{O} + \text{e}^- \rightarrow \text{H}_{2(\text{g})} + \text{OH}^-_{(\text{aq})}$) (*Angew. Chem. Int. Ed.* 2021, 60, 18981–19006). Thus, due to their different catalytic mechanisms, a single type of atomic active site is difficult to boost every elementary reaction of the alkaline HER process, resulting in an insufficient catalytic activity. Given this, it is highly necessary to design HER electrocatalysts containing multiple active sites, which could accelerate the whole alkaline HER process (Volmer–Tafel, Volmer–Heyrovsky or both).

In view of this, we introduced the co-existence of Pt SAs and Pt NPs on the transition metal dichalcogenides (TMDs) of WS_2 substrate by a facile electrodeposition method, which was rarely focused. The content of Pt NPs and Pt SAs can be precisely controlled during the preparation process. Especially, compared with conventional carbon substrate for NPs/SAs deposition (*Angew. Chem.* 2021, 133, 16180–16186; *Adv. Funct. Mater.* 2023, 2213058), the electronic structure of single-atom/clusters metals supported on TMDs is usually adjusted by both the anchoring atom and the neighboring transition metal atoms with relatively high atomic number, which affords a more flexible and complex coordination environment to regulate the catalytic activity. Owing to the various well-defined band structures of TMDs (e.g., WS_2), the core anchoring chalcogen (S) and the neighboring transition metal (W) can synergistically regulate the electronic structure of Pt NPs/SAs through the electronic metal–support interaction. The tunable *d*-orbital state of Pt changes the adsorption energy of reactants on metal atoms and thus influences the catalytic activity of HER (*Nat. Commun.* 2021, 12, 3021). Based on the above conception, the in-depth understanding of synergistic effect on Pt SAs, Pt NCs, and WS_2 substrate were systemically investigated by structural characterizations (XPS, XANES, EXAFS etc.), control experiments design, and DFT calculations in our work. The results deciphered the synergistic effect in $\text{WS}_2/\text{Pt}_{1,n}$ for promoting Volmer–Tafel kinetic rate during alkaline HER, the water dissociation kinetic of Volmer step could be promoted on top site of Pt NC, and the obtained H^* were spilled and immigrated to adjacent S sites of WS_2 for hydrogen production (Tafel step) through fast kinetic rate of Volmer–Tafel pathway, while the Pt single atoms acted as local electron regulator for more thermoneutral H^* adsorption on S active sites. To better embody the conception of the designed $\text{WS}_2/\text{Pt}_{1,n}$ catalysts, some of the above discussion was added to the Introduction section of the revised manuscript.

As for another important part, the azo reaction coupled H_2 production was proposed in our work and realized low-energy-consumption H_2 production compared with traditional OWS. Meanwhile, green electrochemical pathway was achieved to synthesize DAAzF energetic materials, which avoided the traditional hazardous synthetic process. The detailed discussion about azo reaction was displayed in the response of the last Comment.

Comment #2

The authors attempted to introduce a valuable technology for hydrogen production coupled with electrosynthesis. The whole work is talking about reactions at 10 mA cm^{-2} and the overpotential at this

current density (η_{10}). For industry, such low current density is not even close to any useful application. PEM and AEM water electrolyzers work at 1 A cm^{-2} and above, while the alkaline water electrolyzers work at 100 mA cm^{-2} minimum. The η_{10} is meant for comparing model catalysts such as single crystals. For high surface area catalysts, comparing η_{10} does not provide any useful information. Additionally, the energy consumption at such low current density means nothing (Line 25, 26, and 504). The benchmark is not reasonable, as the state-of-the-art water electrolyzer offers at least 1 A cm^{-2} at 1.7 V , not 10 mA cm^{-2} .

Response:

Thanks for your question. Generally, the overpotential (η) at the geometric current density (j) of 10 mA cm^{-2} (η_{10}) is selected as a benchmark for reporting the apparent activity of a catalyst (*Nat. Commun.* 2023, 14, 2460; *Nat. Commun.* 2023, 14, 2306; *Nat. Commun.* 2022, 13, 5843; *Nat. Commun.* 2022, 13, 6863 etc.). In solar water splitting, the potential at 10 mA cm^{-2} is a common figure of merit corresponding to a solar-to-hydrogen efficiency of 12.3%, which efficiency is required for cost-competitive photoelectrochemical water splitting devices (*Science* 2014, 345, 1593–1596; *Chem. Rev.* 2010, 110, 6446–6473; *Chem. Soc. Rev.*, 2015, 44, 5148–5180; *Chem. Soc. Rev.*, 2016, 45, 1529–1541). Indeed, the η_{10} calculated based on the geometric area of electrode cannot represent the intrinsic activity of a catalyst, because the actual surface area of the catalyst that is accessible to the electrolyte varies significantly among different catalysts. Therefore, evaluation of the intrinsic activity of electrocatalysts should be brought to the front (*ACS Energy Lett.* 2019, 4, 1260–1264). For high surface area catalysts, the specific activity obtained by normalizing the current with the actual electrochemical surface area (ECSA) could reflect the intrinsic activity of the catalysts (*J. Am. Chem. Soc.* 2015, 137, 4347–4357). The ECSA-normalized specific activity has been provided in Supplementary Fig. 13 in the initial supporting information. The CC@WS₂/Pt_{1,n} still showed the higher specific activity with a low overpotential of 70.2 mV than that of commercial CC@Pt/C (105.6 mV) at $10 \mu\text{A cm}^{-2}$. To evaluate the high-current-density efficiency and stability of the CC@WS₂/Pt_{1,n}. We have compared the overpotential at 100 mA cm^{-2} (η_{100}) of CC@WS₂/Pt_{1,n} with the reported state-of-the-art WS₂ or Pt-based catalysts (Supplementary Table 2, 3 in the revised supporting information). For a more clearly comparison, a bar graph was performed and shown in Fig. R4. This Figure was added to the Supplementary Fig. 9 in the revised supporting information.

The long-term chronopotentiometry (CP) test of CC@WS₂/Pt_{1,n} at 100 mA cm^{-2} was also carried out (Fig. R5). The results indicated that the CC@WS₂/Pt_{1,n} still deliver high-efficiency and long-stability alkaline HER under high current density because of the in-situ growth of binder-free CC@WS₂/Pt_{1,n} catalysts that can accelerate facile charge transfer and effective mass diffusion, promote the high utilization of exposed active sites, enhance reaction kinetics, and stability, thus producing efficient catalysts (*Adv. Energy Mater.* 2022, 12, 2200409). This section was added to the Supplementary Fig. 18 and Table 3 in the revised supporting information. To evaluate the energy conversion efficiency of our coupling hydrogen production system, the energy input for conventional water electrolysis and coupling system with various electrodes at the target current densities were performed (Fig. R6) (*Nat. Catal.* 2022, 5, 66–73; *Angew. Chem. Int. Ed.* 2022, 61, e202115636; *Adv. Mater.* 2021, 33, 2104791). Because of its low voltage requirement, our coupling hydrogen production system only requires an electricity input of 3.01, 3.35, and 3.70 kWh per m³ of H₂ at the current density of 10, 50, and 100 mA cm^{-2} , respectively, which was lower than conventional water electrolyzers at

the set-up current densities (~ 5.0 kWh per m^3 of H_2) (*Nat. Catal.* 2022, 5, 66–73; *Nat. Energy* 2020, 5, 378). These results suggest that our DAFOR-assisted hydrogen production coupling system may be a promising candidate for energy-efficient hydrogen production. The electricity input analysis was updated to Fig. 7h in the revised manuscript.

Fig. R4. Comparison of the η_{100} of $\text{CC@WS}_2/\text{Pt}_{1,n}$ with the reported state-of-the-art WS_2 or Pt-based catalysts.

Fig. R5. Long-term chronopotentiometry test of $\text{CC@WS}_2/\text{Pt}_{1,n}$ at a current density of 100 mA cm^{-2} .

Fig. R6. Electricity consumed for hydrogen production in conventional OWS and our coupling system using various catalysts.

Comment #3

HER part, the experiment was not correctly done. The Tafel slope of Pt/C should be 120 mV dec^{-1} . Therefore, all the analysis based on the problematic experiment is certainly problematic. Pt/C should be very stable for HER, and thus the results shown in Fig.4j is questionable. Again, η_{10} is only useful for well-defined catalysts. Therefore, based on presented results, I cannot come out the conclusion that $\text{CC@WS}_2/\text{Pt}_{1,n}$ is an advanced HER electrocatalyst.

Response:

Thank you for the question. Tafel slopes which primarily indicates the mechanism of HER and rate determining step (RDS). Since water dissociation coupled proton discharge is the RDS (*i.e.*, Volmer step) of HER in alkaline medium and the Tafel slope of this step should in the range of 50–120 mV dec⁻¹. Any electrocatalyst that is having a Tafel slope ≤ 50 mV dec⁻¹ can be considered a better active electrocatalytic interface (*Chem. Rev.* 2020, 120, 851–918; *Angew. Chem. Int. Ed.* 2021, 60, 18981–19006). Benchmark Pt is the most efficient electrocatalyst for HER with “quasi-zero” onset overpotential together and a small Tafel slope (~ 30 mV dec⁻¹) to deliver 30 mV overpotential at a current density of 10 mA cm⁻² based on the Butler–Volmer equation (*Chem. Soc. Rev.* 2014, 43, 6555–6569).

It is noted that the above Tafel slope are calculated in the low current density region (1–10 mA cm⁻²), while in high current density regions, the Tafel slope of benchmark Pt is closed to 120 mV dec⁻¹ (*Nat. Commun.* 2020, 11, 1116). Therefore, the Tafel slopes of ~ 30 and ~ 120 mV dec⁻¹ for benchmark Pt are all correct that depends on the range of current densities. The Tafel slope of Pt/C in our work was calculated to be 28 mV dec⁻¹, which was closed to the reported literatures (*Nat. Commun.* 2022, 13, 6863 (30.1 mV dec⁻¹); *Nat. Commun.* 2022, 13, 5843 (30 mV dec⁻¹); *Nat. Commun.* 2022, 13, 4200 (29 mV dec⁻¹) *etc.*).

Pt/C is known as the benchmark catalyst for HER, exhibiting a very low overpotential and fast kinetics in acid medium due to its favorable hydrogen adsorption energy. However, the HER kinetics on Pt in the alkaline condition is markedly slower, with the HER rate orders of magnitude lower than that in the acidic electrolyte due to the sluggish water dissociation (Volmer step) and the poor proton supply rate (*Nat. Mater.* 2023, 22, 1022–1029). In addition, the stability of Pt/C under alkaline conditions is also relatively unsatisfactory, with Pt nanoparticles undergoing swift agglomeration on the carbon surface (*Adv. Mater.* 2023, DOI: 10.1002/adma.202303030; *J. Mater. Chem. A*, 2023, 11, 5328–5336). Moreover, the Pt/C with non-3D morphology exhibits several unfavorable reactions which block charge transfer through the increased resistance of the electrocatalyst–substrate interface, reduce the number of active sites and limits the mass transfer rate; which results reduced stability during long-term operation (*Adv. Energy Mater.* 2022, 12, 2200409). All of these factors lead to the sluggish kinetics and unsatisfactory stability during long-term alkaline HER of benchmark Pt/C catalysts. Such phenomenon was in agreement to the recently reported literatures (*Nat. Mater.* 2023, 22, 1022–1029; *Nat. Commun.* 2023, 14, 2460; *Nat. Commun.* 2018, 9, 4958; *Adv. Mater.* 2022, 34, 2206368; *Energy Environ. Sci.*, 2022, 15, 102–108). In contrast, our work introduced the co-existence of Pt NPs and Pt SAs on 3D-structured WS₂ NSs and showed robust stability for at least 100 h to achieve 10 and 100 mA cm⁻² (Fig. 4i, 4k and Fig. R5). On the other hand, experimental design and DFT calculations revealed that the water dissociation kinetic of Volmer step could be promoted on top site of Pt NC in CC@WS₂/Pt_{1,n}, and the obtained H* were spilled and immigrated to adjacent S sites of WS₂ for hydrogen production (Tafel step) through fast kinetic rate of Volmer–Tafel pathway, while the Pt single atoms acted as local electron regulator for more thermoneutral H* adsorption on S active sites. As a result, the CC@WS₂/Pt_{1,n} showed low overpotentials of 27.1 and 60.4 mV to achieve 10 and 100 mA cm⁻² for alkaline HER (the current density of 10 mA cm⁻² is equivalent to 12.3% efficiency of a solar water-splitting device, so the overpotential needed for this current density can provide a means for comparison. η_{10} is usually used to compare the activities of various catalysts in HER, which has been discussed in Comment #2) (*Chem. Rev.* 2020, 120, 851–918; *Chem. Soc. Rev.*,

2014, 43, 6555–6569). The highly-efficient CC@WS₂/Pt_{1,n} for alkaline HER were superior to the benchmark Pt/C and majority of recently-reported Pt- or WS₂ based catalysts (Fig. 4c and Fig. R4). Based on the above discussion, the elaborate CC@WS₂/Pt_{1,n} is prove to be an advanced alkaline HER electrocatalyst.

Comment #4

Line 444, I am not sure about that the control experiment for the radical mechanism. CuO is a poor catalyst for isopropanol electrooxidation, and therefore isopropanol may be a poison for CuO which may also cause the reduced current.

Response:

Thank you for pointing this out. As you proposed, the active sites of CuO electrode may be blocked by the adsorption of isopropanol due to the low activity for isopropanol electrooxidation reaction (IOR), which may be the main reason for the reduced current rather than the decreased quantity of hydroxyl radicals. To improve the IOR efficiency, the Pt film electrode was used in the control experiment alternative to CuO electrode because the state-of-the-art alcohol oxidation reaction electrocatalysts are mainly based on noble metals, such as Pt, Pd, and Rh, owing to their superior properties towards adsorption and dehydrogenation of alcohol (*Nat. Catal.* 2019, 2, 495–503; *Angew. Chem. Int. Ed.* 2023, 62, e202305158; *Sci. Bull.*, 2021, 66, 2079–2089). As shown in Fig. R7, a similar decrease in current response was observed in the DAF aqueous electrolyte when isopropanol was added as the •OH scavenger. Taken together with the control experiment for DAFOR with the addition of TEMPO •OH scavenger, the results indicated that the oxidative-coupling of DAF into DAAzF EMs was likely driven by hydroxyl radicals. This section was updated to Supplementary Fig. 26 in the revised supporting information.

Fig. R7. LSV curves of DAFOR in 0.20 M DAF+1.0 M KOH with or without the addition of isopropanol free radical scavengers on Pt film.

Comment #5

Again, reactions (HER, DAFOR, and coupled one) at 10 mA cm^{-2} is not of interest to industry. The authors have to present results at high current density to showcase the practical value of their system. Additionally, increasing energy efficiency by reducing the operation voltage (or, current density) is nonsense.

Response:

Thank you for the suggestion. As suggested by the reviewer, in addition to the working potential at 10 mA cm⁻², the values at higher current density (eg. 100 mA cm⁻²) were also presented for alkaline HER, DAFOR, and coupling system (Fig. R4, Fig. 6e, and Fig. 7b, c in the revised manuscript).

For alkaline HER, the CC@WS₂/Pt_{1,n} showed low overpotential of 60.4 mV to achieve 100 mA cm⁻², outperforming the Pt/C (81.9 mV) and majority of the recently reported Pt- and WS₂-based HER catalysts (Fig. 4c, Fig. R4). For DAFOR process, the LSV curves of reaction at various concentrations of DAF were carried out, the corresponding working potentials at 10 and 100 mA cm⁻² were also presented (Fig. 6e). The results showed that the DAF concentrations of 0.2 M delivered the lowest working potentials of 1.23 and 1.44 V vs. RHE, which were still much lower than those of benchmark OER catalysts (*Nat. Commun.* 2023, 14, 1412; *Nat. Commun.* 2023, 14, 1792; *Nat. Commun.* 2023, 14, 1248), indicating that the DAFOR was the potential alternative to OER. The coupling system of HER||DAFOR using CC@WS₂/Pt_{1,n} and CF@CuO as cathode and anode electrodes respectively showed the cell voltages of 1.26 and 1.55 V to achieve 10 and 100 mA cm⁻², far lower than the values for conventional OWS system (1.67 and 1.99 V), revealing the potential practical value of our coupling system (Fig. 7b).

In some cases, a higher hydrogen production rate is needed and required a larger operating current density, which will ultimately increase the electricity consumption and thereby decrease the efficiency of hydrogen production per unit. Thus, it is practical to increase the yield of hydrogen gas at the same current density. The electricity consumption per m³ of H₂ produced (W, kWh per m³ H₂) was calculated as: $W = (n \times F \times U \times I, 000)/(3,600 \times V_m)$, where n is the number of electrons transferred for each product molecule ($n=2$ for conventional water electrolysis and coupling system), U is the applied cell voltage of the coupling system, V_m is the molar volume of gas at normal temperature and pressure (22.4 mol L⁻¹). It can be seen that the electricity consumption is directly proportional to the applied cell voltage, suggesting that the change of cell voltage could evaluate the energy conversion efficiency of the coupling system for hydrogen production (*Nat. Catal.* 2022, 5, 66–73; *Nat. Catal.* 2023, 6, 218–219; *Angew. Chem. Int. Ed.* 2023, 62, e202214333; *Angew. Chem. Int. Ed.* 2022, 61, e202213328). For a better understanding, the calculation details of electricity consumption were added to the Methods section in the revised manuscript. The calculated result was presented in the Page 26 in the revised manuscript.

Comment #6

Fig. 7d is useless but misleading. If HER coupled with alcohol oxidation, the overall voltage can be reduced to a value way below 1 V. The point of decoupled HER/OER is not at how low voltage you can produce hydrogen, it is how much value you can add by the coupled reaction.

Response:

Thanks for your question. Indeed, the coupling of HER and alcohol oxidation reaction (AOR) may lower the cell voltage below 1 V for more energy-saving hydrogen production due to the thermodynamically more favorable potential of AOR (eg. methanol (0.103 V), ethanol (0.057 V), and glycerol (0.003 V) vs. RHE) than that of OER (*Nat. Commun.*, 2023, 14, 1686; *J. Am. Chem. Soc.*, 2022, 144, 7224–7235; *Adv. Energy Mater.*, 2022, 12, 2201047). However, it is hard to compare cell voltage of various small-molecule oxidation reactions coupled OWS system because of the different

thermodynamic oxidation potential of the small-molecule oxidation reactions. Because a new substitute of DAFOR to OER was involved in the OWS in our work, it is meaningful to compare the cell voltage of our coupling system with the conventional OWS to illustrate the thermodynamically more favorable of DAFOR than OER and achieves low cell voltage of the designed novel coupling system for energy-efficient hydrogen production.

Conventional OWS involves a sluggish four-electron OER reaction, resulting in hydrogen production at very high cell voltage and energy consumption. Replacing the OER reaction with thermodynamically more favorable oxidation reaction can effectively reduce the cell voltage for generating clean H₂ with less energy input. In addition, value-added organic compounds can be generated on the anode through electrooxidation upgrading (*Adv. Energy Mater.* 2022, 12, 2201047; *Angew. Chem. Int. Ed.* 2022, 61, e202213328; *Angew. Chem. Int. Ed.* 2022, 61, e202209014). Therefore, the change in cell voltage for hydrogen production on the cathode and value-added chemicals generation on the anode side should both be taken into account. The new oxidation reaction of DAFOR with thermodynamically more favorable than OER was introduced to OWS in our work achieved low-cell-voltage hydrogen production with lower energy input compared with conventional OWS. Meanwhile, value-added DAAzF EMs rather than oxygen was generated at the anode side, which avoids the drawbacks brought by oxygen such as forming explosive H₂/O₂ gas mixtures. Our work also introduced a green electrosynthesis pathway to synthesize DAAzF EMs instead of the traditional hazardous synthetic condition (high-temperature, requirement of chemical oxidant and by-products separation process). As a result, the well-designed novel coupling system in our work achieved co-production of valuable chemicals of H₂ and DAAzF EMs through a green and energy-saving pathway.

Comments:

Overall, the work has potential, and the story could be much more exciting by focusing on the azo reaction.

Reply:

Thank you for your suggestion on the azo reaction. As discussed in the Comment #1, apart from the construction of novel coupling system assembled by new anodic azo reaction and cathodic HER proposed in our work, electrocatalysts play a decisive role in controlling the electricity consumption and energy conversion efficiency of water electrolysis systems; therefore, their design and synthesis is extremely important. Catalytic electrodes, integrating the conductive substrate and electrocatalysts into a single body without any additional binder, could become more popular on exhibiting low cost, abundant reserves, robustness, high activity, large size, specific surface area, and simple preparation (*Nano Energy* 2022, 104, 107875; *Adv. Energy Mater.* 2022, 12, 2200409). Our work introduced the co-existence of the Pt NPs, Pt SAs loaded on the WS₂ substrate, which was fundamentally different from the traditional SACs and offered some advantages such as thermodynamically more stable feature, highly-efficient activity for multi-step alkaline HER by the synergy among various active centers of Pt NPs, Pt SAs, and S sites of WS₂ substrate. The structural characterization, alkaline HER tests, and theoretical calculations confirmed the synergistic effect of Pt NPs, Pt SAs, and WS₂ that were responsible for promoting the efficiency and stability of the CC@WS₂/Pt_{1,n} catalysts.

As for another important part of azo reaction on the anode, the optimal concentration of DAF

starting material were firstly investigated (Fig. 6d, e). Then, the adsorption capacity of DAF on the surface of various catalysts was discussed *via* DFT calculations that combined with LSV tests to select suitable catalyst electrode (Fig. 6f and Supplementary Fig. 25). The DAFOR mechanism was then investigated by the control experiments through the addition of radical scavenger to reveal that the $\bullet\text{OH}$ free radical was acted as oxidant for oxidative-coupling of DAF into DAAzF EMs (Supplementary Fig. 26). The successfully produced DAAzF EMs on the anodic electrolyte was confirmed by ^{13}C NMR and FTIR. The high-energy feature of DAAzF was supported by DSC measurement (Fig. 7g and Supplementary Fig. 28, 30).

To put forward the story of DAFOR, the Raman spectra was further supplemented in the revised manuscript to eliminate the influence of water molecules in FTIR and further confirmed the successful transform of DAF to DAAzF EMs *via* N–N oxidative-coupling mechanism. As shown in Fig. R8, the formation of the azo compounds is confirmed by comparing the Raman spectra of the reaction products with that of the starting material. Compared with initial DAF alkaline solution, two dominate peaks located at 1295 and 1428 cm^{-1} were detected after CP test, corresponding to the C–N_{azo} stretching and the stretching mode of the azo group, respectively (*Combust. Flame*, 2022, 240, 111981; *Chem. Mater.* 2008, 20, 1750–1763; *Chem. Mater.* 2005, 17, 3784–3793). In addition, the decreased intensity of $\gamma(\text{NH}_3)$ and $\omega(\text{C–NH}_2)$ stretching modes indicated the dehydrogenation of amino group and then involved N–N oxidative-coupling process to form DAAzF EMs (*J. Phys. Chem. C* 2019, 123, 8731–8739). Taken together with the ^{13}C NMR and FTIR analysis, the anodic product could be identified to the pure DAAzF EMs through the N–N oxidative-coupling of DAF starting material. This section was added to the Supplementary Fig. 29 in the revised supporting information.

Fig. R8. Raman spectra of anode electrolyte before and after long-term CP test of coupling system, the Raman spectra of pure 1.0 M KOH and air were set for comparison (inset: Raman spectra at lower frequency).

REVIEWER COMMENTS

Reviewer #1 (Remarks to the Author):

The authors have provided appropriate revision according to the reviewers' comments with additional experiments and related discussions, which can further enhance the quality of the manuscript. The reviewer thus suggests the acceptance of the revised version of this work.

Reviewer #2 (Remarks to the Author):

In fact, it is not reasonable to simulate Pt NPs with Pt13, but we also know that calculating clusters like Pt55 or larger clusters is difficult. However, using the stable structure of the cluster as the initial structure in the process of optimizing the structure of the cluster on the surface is a separate issue from ensuring that the form of Pt13 is stable on the surface. You may need to do AIMD, and if this system is in the solution phase, you should use a program like VASPsol.

The G_H values are -2.7 on Pt NC, which seems too low. This indicates that Ptn cluster itself cannot be used as catalytic active form instead PtmHn cluster may be working [ACS Nano 2009, 3, 7, 1657-1662]. Therefore, your model lacks of reality.

The G_H value for pristine WS2 is too low. Because you included the G_H values for edge of pristine WS2. The G_H value for the basal plane S is more than 2.0 eV.

You included the reference [Nat. Commun. 2022, 13, 3063], but the paper is not WS2 paper.

Overall, It is difficult to say that the current calculation results actually simulate the actual situation and are of little help. There is a lot of potential for readers to misunderstand.

Reviewer #3 (Remarks to the Author):

I appreciate the authors' effort for the revision. Almost all my comments have been addressed. There is only one left, but it is really a big one. Therefore, it should be corrected before publishing. After that, it is good to go.

Tafel analysis. The authors were incorrect on it. Since, the mechanism has been discussed and referred to DFT calculation, it is very important to make it right. Their response revealed the problem. The mistake may involve: 1) wrong region, 2) wrong experiment, i.e. the test should be taken in H2-saturated electrolyte. The authors quoted from a few high-profile publications. Unfortunately, high-profile does not mean correct. A good reference is that for Pt the Tafel slope is ~120, not ~39 mV/dec. Please see Wenchao Sheng's recent paper for help, as well as old literature from Hubert Gasteiger.

Dear Editor:

Much thanks for all the comments from all the reviewers, which helps us to improve our manuscript a lot. We listed the answers in the following text.

Response to Reviewer #1:

Comments:

The authors have provided appropriate revision according to the reviewers' comments with additional experiments and related discussions, which can further enhance the quality of the manuscript. The reviewer thus suggests the acceptance of the revised version of this work.

Reply:

We express our sincere gratitude to the reviewer for your careful review on our manuscript.

Response to Reviewer #2:

Comments:

In fact, it is not reasonable to simulate Pt NPs with Pt13, but we also know that calculating clusters like Pt55 or larger clusters is difficult. However, using the stable structure of the cluster as the initial structure in the process of optimizing the structure of the cluster on the surface is a separate issue from ensuring that the form of Pt13 is stable on the surface. You may need to do AIMD, and if this system is in the solution phase, you should use a program like VASPsol.

Response:

We appreciated for this constructive comment, which really helps to improve the quality of our manuscript. Following the suggestions, we asked some experts for help to include the results of *ab initio* molecular dynamics (AIMD) simulations using VASP code as a complement to evaluate the dynamical stability of the Pt₁₃ clusters on WS₂. The AIMD calculations were run under an NVT ensemble at T = 350 K for t = 1 ps total simulation time for WS₂/Pt₁₃ (Fig. R1). From the optimized structure, no obvious changes in geometries or energies were observed, evidencing the good stability of the WS₂/Pt₁₃ composite, this section was added to the Fig. S19 in the revised supplementary information. In addition, the effects of implicit solvation were considered using VASPsol software to recalculate ΔG_{H^*} at several adsorption sites for WS₂/Pt₁₃, WS₂/Pt SA, and pristine WS₂ (see below). This section was updated to Page 20 in the revised manuscript. We hope this revised manuscript would satisfy the reviewer.

Fig. R1. AIMD simulation trajectories as well as the obtained optimal structures for WS₂/Pt₁₃.

Comment #1

The G_H values are -2.7 on Pt NC, which seems too low. This indicates that Ptn cluster itself cannot be used as catalytic active form instead PtmHn cluster may be working [ACS Nano 2009, 3, 7, 1657–1662]. Therefore, your model lacks of reality.

Response:

Yes, this question has some relation with the above comment. Based on the result of AIMD (Fig. R1), the unusual ΔG_{H^*} value on Pt₁₃ may not due to the unstable structure of Pt₁₃ on the WS₂. We then recalculated the ΔG_{H^*} at several adsorption sites for WS₂/Pt₁₃ via VASP code (*Phys. Rev. B* 54, 11169–11186 (1996)). The polarizable implicit solvent models were considered for simulating the H₂O solvent environment and the dielectric constant was set to 80 as implemented in the VASPsol (*J. Chem. Phys.* 2014, 140, 084106). Under the effect of implicit solvation, the ΔG_{H^*} of WS₂/Pt₁₃ at the top and side sites were -0.416 and -0.463 eV, respectively (Fig. R2). These results were closed to the reported Pt₁₃ cluster-based literatures (*Adv. Energy Mater.* 2023, 2204213; *J. Mater. Chem. A*, 2021, 9, 5468–5474; *J. Catal.* 2019, 375, 351–360). This section was updated to Fig. 5i in the revised manuscript and Fig. S19 in the revised supplementary information.

Fig. R2. Calculated ΔG_{H^*} on different sites of WS₂, WS₂/Pt SA, and WS₂/Pt NC.

Comment #2

The G_H value for pristine WS₂ is too low. Because you included the G_H values for edge of pristine WS₂. The G_H value for the basal plane S is more than 2.0 eV.

Response:

Thanks for your question. The ΔG_{H^*} of pristine WS₂ at the basal plane of S site was recalculated with the effects of implicit solvation *via* VASPsol (Fig. R2). The result showed that the ΔG_{H^*} of basal plane of S site on pristine WS₂ was 2.069 eV, indicating the inert basal plane of WS₂ that lead to the weak adsorption of H*. This phenomenon agreed well with the reported literatures (*Energy Environ. Sci.*, 2018, 11, 2270–2277; *Adv. Mater.* 2020, 32, 2002584; *Adv. Funct. Mater.* 2022, 2112362; *J. Catal.* 2020, 382, 204–211). This section was updated to Fig. 5i in the revised manuscript and Fig. S20 in the revised supplementary information. The summary of ΔG_{H^*} on various adsorption sites of reported pristine WS₂ structures were also updated in supplementary Table 4. For a better comparison, the ΔG_{H^*} of WS₂/Pt SA at the Pt SA and S sites were also recalculated *via* VASP code with the effects of implicit solvation *via* VASPsol (Fig. R2). The results showed that the Pt SA site of WS₂/Pt SA exhibited the most thermoneutrality for H* adsorption with the ΔG_{H^*} value of only –0.05 eV, much lower than those of Pt NC, WS₂/Pt SA-S site (1.648 eV), and pristine WS₂-S site. The discussion of ΔG_{H^*} on various sites of pristine WS₂, WS₂/Pt SA, and WS₂/Pt NC were updated to Page 20 in the revised manuscript.

Comment #3

You included the reference [Nat. Commun. 2022, 13, 3063], but the paper is not WS₂ paper.

Response:

Yes. We have checked the literature carefully and the improper citation was corrected, a WS₂-related reference was updated to Ref. 36 (*Nat. Commun.* 2021, 12, 3021) in the revised manuscript. Thanks for your kind reminding.

Comments

Overall, It is difficult to say that the current calculation results actually simulate the actual situation and are of little help. There is a lot of potential for readers to misunderstand.

Reply:

As for the theoretical calculation, according to all the comments related to this, the AIMD simulation was performed to evidence the good stability of WS₂/Pt₁₃. Thereafter, the ΔG_{H^*} on various adsorption sites for WS₂/Pt₁₃, WS₂/Pt SA, and pristine WS₂ were recalculated *via* VASP code. The effects of implicit solvation were considered using VASPsol software to simulate the actual situation of HER in aqueous solution during the calculation. Much thanks for the help of the theoretical investigation from Prof. Bingbing Suo and Haiyan Zhu in Northwest University. We hope the revised part of the theoretical calculations would provide a clear interpretation for readers to understand the intrinsic interaction between Pt_{1,n} and WS₂ substrate in promoting alkaline HER activity.

Response to Reviewer #3:

Comments:

I appreciate the authors' effort for the revision. Almost all my comments have been addressed. There is only one left, but it is really a big one. Therefore, it should be corrected before publishing. After that, it is good to go.

Reply:

We express our sincere gratitude to the reviewer for your careful review on our manuscript and the constructive comments, which really help to improve the quality of our manuscript. Following the reviewer's suggestions, the Tafel slopes of Pt/C and prepared HER catalysts were re-evaluated and the possible alkaline HER mechanism was proposed. We hope this revised manuscript would satisfy the reviewer.

Comment #1

Tafel analysis. The authors were incorrect on it. Since, the mechanism has been discussed and referred to DFT calculation, it is very important to make it right. Their response revealed the problem. The mistake may involve: 1) wrong region, 2) wrong experiment, i.e. the test should be taken in H₂-saturated electrolyte. The authors quoted from a few high-profile publications. Unfortunately, high-profile does not mean correct. A good reference is that for Pt the Tafel slope is ~120, not ~39 mV/dec. Please see Wenchao Sheng's recent paper for help, as well as old literature from Hubert Gasteiger.

Response:

Thank you for the suggestion. Indeed, the Tafel tests are preferred to be performed with H₂ saturation, because a well-defined equilibrium potential of 0 V (vs the reversible hydrogen electrode, RHE) for HER can only be established under such condition (Sheng *et al. ACS Catal.* 2023, 13, 2534–2541; Sheng *et al. Adv. Mater.* 2019, 31, 1808066). In addition, 30 mV dec⁻¹ has often been reported for HER on Pt, which is caused by misconducting kinetic analysis in a non-Tafel regime. Tafel analysis should be made in strong polarization regime, where the reverse reaction proceeds at sufficiently slow rate and thus can be neglected. Consequently, the Tafel regime is typically beyond –59 mV for HER as determined by the Butler–Volmer equation (Sheng *et al. Adv. Mater.* 2019, 31, 1808066; Gasteiger *et al. J. Electrochem. Soc.* 2010, 157, B1529-B1536). In view of this, the Tafel slopes of CC@WS₂/Pt_{1,n} and CC@Pt/C were evaluated in H₂-saturated 1.0 M KOH and shown in Fig. R3. The Tafel slope of CC@WS₂/Pt_{1,n} was calculated as 27.5 mV dec⁻¹, which was much lower than that of CC@Pt/C (118.5 mV dec⁻¹). Accordingly, the possible alkaline HER mechanism was proposed to Volmer–Tafel pathway with Tafel step as the rate-determining step (RDS) (Sheng *et al. Adv. Mater.* 2019, 31, 1808066). This section was updated to Fig. 4d in the revised manuscript.

Fig. R3. Tafel plots of CC@WS₂/Pt_{1,n} and CC@Pt/C.

REVIEWERS' COMMENTS

Reviewer #2 (Remarks to the Author):

There were major errors in the calculation results and discussion in the previous version of manuscript, but now the major errors are almost gone.

Reviewer #3 (Remarks to the Author):

good to go